# FROM TRAINING-FREE TO ADAPTIVE: EMPIRICAL INSIGHTS INTO MLLMs' UNDERSTANDING OF DETECTION INFORMATION

## ABSTRACT

Despite the impressive capabilities of Multimodal Large Language Models (MLLMs) in integrating text and image modalities, challenges remain in accurately interpreting detailed visual elements. Fortunately, vision detection models have shown superior performance in recognizing fine-grained image details, leading to their increased deployment by researchers to enhance the ability of MLLMs. Among the feasible strategies, infusing detection information in text format is easy to use and effective. However, most studies apply this method in a training-free manner. There is limited research on the effects of adaptive training, which has great potential for helping LLMs better comprehend the special input and discard irrelevant information. In this paper, we address the key research question: How does training influence MLLMs' understanding of infused textual detection information? We systematically conduct experiments with numerous representative models to explore the performance implications of training-free, retraining, and fine-tuning strategies when infusing textual detection information into MLLMs. Additionally, we investigate the impact of training on the original abilities of MLLMs, as well as the interchangeability of detection models. We find that fine-tuning the pre-trained MLLM to adapt to textual detection information yields better results compared to the training-free strategy and the retraining strategy, with the fine-tuned MLLM outperforms the training-free MLLM by 6.71% across 10 widely recognized benchmarks. Besides, we find that fine-tuning allows the MLLM to maintain performance improvements even after replacing the deployed detection models, which means that it enables the MLLM to better understand the specially formatted textual information. We release our codes to facilitate further exploration into the fusion strategies of vision detection models and improving the fine-grained multimodal capabilities of MLLMs.

## 1 INTRODUCTION

The advent of large language models (LLMs) has marked a transformative era in natural language processing (Brown et al., 2020; Touvron et al., 2023), paving the way for the development of Multimodal Large Language Models (MLLMs) that blend linguistic and visual understanding. Pioneers such as GPT-4V have demonstrated remarkable proficiency across numerous tasks (Yang et al., 2023). However, a notable gap remains in these models' ability to accurately discern and recognize fine details within images (Fu et al., 2023). This limitation is particularly evident when MLLMs generate coherent yet misaligned responses with the image content, a phenomenon often referred to as "hallucination" (Li et al., 2023b; Huang et al., 2023).

Current advancements in object detection and optical character recognition (OCR) models have established their effectiveness in identifying objects and text within images (Zou et al., 2023; Liu et al., 2024b). Consequently, researchers have increasingly deployed vision detection models to assist MLLMs in recognizing fine-grained visual elements. A popular approach involves converting the outputs of vision detection models into textual descriptions, which are then supplied to the backbone LLM, thereby enhancing the MLLM's performance in visual tasks. This fusion strategy is both straightforward and effective.

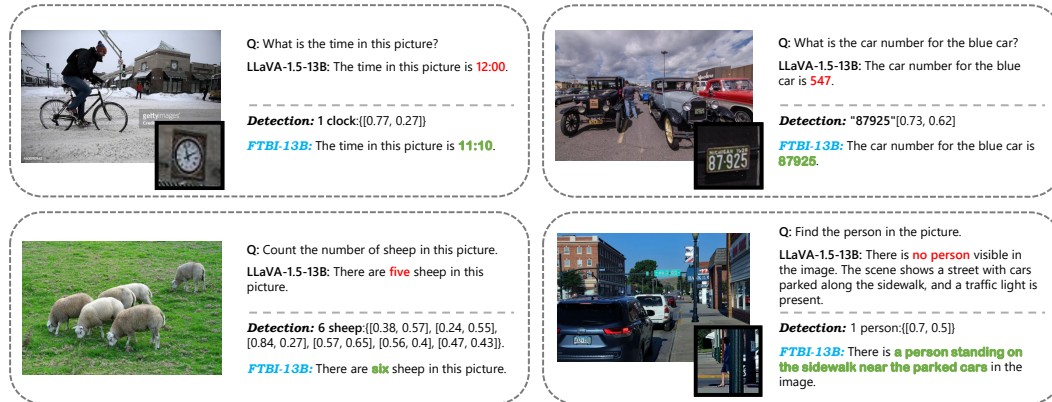

Figure 1: Examples where LLaVA-1.5-13B fails, while the model infused with textual detection information (FTBI-13B) succeeds. "*Detection*" refers to processed detection information from OD/OCR models. Additional examples are provided in Figure 5 of Appendix A.1.

Nonetheless, the majority of existing research has primarily focused on training-free methods to directly apply the textual detection information [1]. Little exploration has been conducted into adaptive training methods, which have great potential to enhance LLMs' comprehension of specially formatted textual content, enabling them to intentionally discard irrelevant information and generate more pertinent responses (Zhang et al., 2024b; Cabessa et al., 2024). This highlights the need for a systematic investigation, particularly concerning the core research question: **Can adaptive training further enhance MLLMs' performance beyond what is achievable through training-free integration of textual detection information?**

To provide insights into how training impacts the infusion of textual detection information into MLLMs, we investigate training-free, retraining, and fine-tuning strategies for this fusion method. Additionally, we examine how training influences the original image understanding capabilities of MLLMs and the interchangeability of deployed detection models. Based on the experimental analysis encompassing representative advanced models, including LLaVA-1.5 (Liu et al., 2023a), DINO (Zhang et al., 2022), PaddleOCRv2 (Du et al., 2021), and Grounding DINO (Liu et al., 2023c), alongside Qwen-VL (Bai et al., 2023) and YOLOv8 (Jocher et al., 2023) in the appendix, we systematically uncover the following key insights:

**(1) The fine-tuning strategy yields better results than both the training-free and retraining strategies.** Building on prior studies (Wu et al., 2024; Wang et al., 2024a; Chen et al., 2024; Zhou et al., 2023), we convert the output of vision detection models into textual information and input it into the LLM. We explore three distinct training strategies: **the training-free strategy**, where detection information is directly fed into the MLLM without additional training; **the retraining strategy**, which involves retraining the MLLM from scratch and continuously infusing textual detection information; and **the fine-tuning strategy**, where additional fine-tuning is applied to a pre-trained MLLM to help it comprehend the specially formatted information. Evaluating performance across ten widely recognized benchmarks, we find that all three strategies enhance LLaVA-1.5's performance in fine-grained image recognition. Notably, the fine-tuning strategy achieves the most significant improvements, elevating performance by up to 6.71% compared to the training-free approach.

**(2) Retraining with textual detection information impairs MLLMs' original image comprehension abilities.** Most advanced MLLMs employ an image encoder to generate image features, and their ability to understand these features is crucial for effective multimodal understanding. Our experiments reveal that retraining the MLLM with textual detection information detrimentally affects its ability to interpret the features from its image encoder. In contrast, the fine-tuning strategy does not run into this problem.

---

[1]To maintain brevity, we refer to "*textual detection information*" as the information output by vision detection models in textual format.

**(3) Fine-tuning allows the MLLM to retain performance improvements upon replacing the deployed detection model.** The characteristics and performance of the deployed detection models significantly influence the enhanced MLLM's effectiveness. Based on the fine-tuning strategy, we examine replacing a closed-set detector with an open-set detector. The results demonstrate further enhancement in MLLM performance, enabling dynamic object detection following the context of user queries during inference. Additionally, we find that the fine-tuned MLLM maintains its training benefits and can still effectively discard irrelevant information even after the model replacement.

To summarize, our work contributes comprehensive empirical evidence and practical insights into the effects of various training strategies for infusing textual detection information into MLLMs. It identifies a significant gap between the use of adaptive training and training-free methods, highlighting the potential of adaptive strategies and demonstrating their feasibility through systematic investigation. Our code is publicly available at *anonymous link* to facilitate further research and pave the way for systems that engage in more nuanced and accurate multimodal dialogue.

## 2 BACKGROUND AND MOTIVATION

### 2.1 MULTIMODAL LARGE LANGUAGE MODELS (MLLMs)

**Linking Text and Vision Information.** Large Language Models (LLMs) are primarily designed for text-based tasks (Zhao et al., 2023). To incorporate image processing capabilities, modality bridging modules have been developed to reconcile the representation differences between text and images (Yin et al., 2023). Generally, these methods can be categorized into three types:

(1) *Learnable queries* are used to distill information from image features. For instance, Flamingo (Alayrac et al., 2022) employs a perceiver resampler, and IDEFICS (Hugo et al., 2023; Laurençon et al., 2024) uses similar modules to extract features from Vision Transformers (ViT) (Dosovitskiy et al., 2020). BLIP-2 (Li et al., 2023c) utilizes learnable queries alongside a Q-Former module, while Qwen-VL (Bai et al., 2023) compresses visual features into sequences of fixed length using cross-attention layers. (2) *Projection-based interfaces* bridge modalities with straightforward techniques. Notable examples include LLaVA (Liu et al., 2023b;a; 2024a) and MGM (Li et al., 2023d), which utilize simple linear layers to map image features into the text semantic space. (3) *Parameter-efficient tuning modules* are utilized to fine-tune MLLMs for image feature comprehension. For example, LLaMA-Adapter (Zhang et al., 2023; Gao et al., 2023) introduces self-attention layers with zero gating for fine-tuning, and LaVIN (Luo et al., 2023) employs modality-specific adapters.

**Why Incorporating Detection Models into MLLMs?** Existing MLLMs often struggle to accurately detect fine-grained targets. For example, in Figure 1, LLaVA-1.5 miscounts a herd of sheep, indicating a limitation in its object-counting capability. Additionally, it fails to detect a pedestrian who is partially obscured by a utility pole, highlighting a weakness in its object localization ability. In another scenario, LLaVA-1.5 incorrectly recognizes the license plate number "87025" as "547", revealing a shortcoming in its text recognition ability. By contrast, SOTA object detection and OCR models demonstrate superior performance on detection and recognition tasks, which has led many researchers to explore the application of detection models within the realm of MLLM research.

### 2.2 ENHANCING DETECTION CAPABILITIES FOR MLLMs

**Existing Methods for Detection Capabilities Enhancement.** Various strategies have been explored to enable MLLMs aware of image details, generally classified into four types:

(1) *Expanding datasets with existing object detection or OCR data:* InstructBLIP (Dai et al., 2023) utilizes data from 26 datasets across 11 tasks, including OCR data. ASM (Wang et al., 2023a) introduces 1 billion region-text pairs. LLaVA and SPHINX (Lin et al., 2023) compile hybrid instruction fine-tuning datasets, incorporating object detection datasets like VG (Krishna et al., 2017) and the OCR dataset OCRVQA (Mishra et al., 2019). PINK (Xuan et al., 2023) employs a bootstrapping method to cover diverse referential comprehension datasets. MiniGPT4-v2 (Chen et al., 2023b), VisionLLM (Wang et al., 2024b), and Shikra (Chen et al., 2023c) integrate object detection datasets, such as RefCOCO (Kazemzadeh et al., 2014), PointQA (Mani et al., 2020), and Flickr30K (Plummer et al., 2015), while introducing special detection tokens like *"det"* to guide downstream tasks (further details in Appendix D.7).

(2) *Restructuring the image encoder to extract fine-grained features*: LION (Chen et al., 2023a) introduces a Vision Aggregator module for feature aggregation, while Honeybee (Cha et al., 2023) employs a deformable attention-based abstractor for capturing fine details. UReader (Ye et al., 2023) utilizes a shape-adaptive cropping module to process local image features, and Vary (Wei et al., 2023b) develops a dedicated image encoder for text recognition. Eagle (Shi et al., 2024) aligns features from various visual experts, concatenating them as input for the MLLM. Mova (Zong et al., 2024) introduces the MoV-Adapter, which extracts and fuses task-specific knowledge.

(3) *Integrating pre-trained detection models into MLLMs' output end to train MLLMs or perform detection tasks*: UNIFIED-IO (Lu et al., 2022; 2023) unifies image, text, and detection features into discrete tokens and trains an end-to-end MLLM capable of detecting. ContextDET (Zang et al., 2023) trains a visual decoder for bounding box prediction using contextual LLM tokens. Lenna (Wei et al., 2023a), Lisa (Lai et al., 2023), and Next-chat (Zhang et al., 2024a) introduce additional tokens to prompt detectors for target identification.

(4) *Converting detection model outputs into text and using it as supplementary input for LLMs*: GLEE (Wu et al., 2024) builds on LISA (Lai et al., 2023) to generate SEG tokens for targeted segmentation, enhancing performance by feeding textual object queries into the backbone LLM. $P^2G$ (Chen et al., 2024), Moai (Lee et al., 2024), and IVE (He et al., 2024) employ detection agents to generate textual grounding clues for improved reasoning. Power-LLaVA (Wang et al., 2024a) utilizes an object detector to produce textual class and location information to assist the MLLM in generating high-quality outputs. VLPrompt (Zhou et al., 2023) leverages an object detector to generate target names and infer relationships, thereby aiding MLLMs in reasoning tasks.

**Why Adaptive Training with Textual Detection Information?** Although methods in the second and third categories can improve the detection capabilities of MLLMs, they typically require substantial datasets to train the restructured image encoders or achieve feature alignment. In contrast, text-based methods are simpler, necessitate less extensive training data for the newly built detection modules, and still deliver commendable results. Thus, the fourth type of method is likely to be more frequently employed in practical applications.

While most research in this category has concentrated on training-free strategies for infusing textual detection information, we note relevant developments in pure-text LLMs. Zhang et al. (2024b) propose leveraging Retrieval-Augmented Generation (RAG, Gao et al. (2024)) during fine-tuning to help LLMs discard redundant information from augmented text. Additionally, Cabessa et al. (2024) suggest that infusing well-crafted textual features during fine-tuning can enhance LLMs' comprehension of specially formatted inputs. They primarily use a small amount of data to adaptively train LLMs for comprehending specially formatted text, yielding excellent results.

This leads us to an important question: *Since the infusion of textual detection information already performs well without training, could this fusion method achieve even better outcomes with appropriate training?* Our work aims to address this question by utilizing the original training data of the studied MLLMs, which is limited in quantity but high in quality, to conduct adaptive training for the infusion of textual detection information into MLLMs.

## 3 INVESTIGATION METHODOLOGY FOR THE INFUSION OF TEXTUAL DETECTION INFORMATION

### 3.1 TEXT-BASED DETECTION INFORMATION CONSTRUCTION

Similar to many studies (Wang et al., 2024a; Chen et al., 2024; Zhou et al., 2023), we first need to convert the output of object detection models and OCR models into specially formatted text.

**Object Detection Information.** With object detection models, we can extract information about class labels and bounding box coordinates of identified objects. We present results using a popular and advanced model, DINO (Zhang et al., 2022), on the main page as a representative. Specifically, we first convert the output of DINO into text. To shorten the sentence, we select the first two values from the bounding box coordinates as positional information, which represent the central coordinates of the objects. Then, we consolidate objects within the same category, further reducing the length while serving as a counter. Finally, we add an instruction sentence before

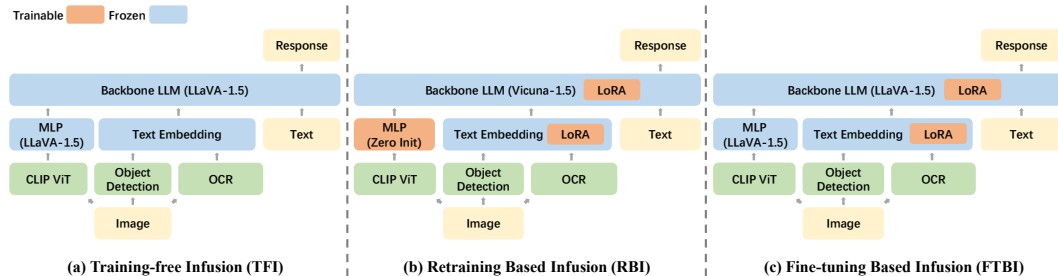

Figure 2: The studied MLLM architectures with different training strategies for infusing textual detection information. "(LLaVA-1.5)" denotes module initialization with weights from LLaVA-1.5.

the category and coordinates information to create the final sentence, which is looks like: "Here are the central coordinates of certain objects in this image: 2 people:{[0.25, 0.12], [0.11, 0.43]}, 1 cake:{[0.42, 0.32]}."

**OCR Information.** With OCR models, we can extract textual content within images along with their positional information. In the main page, we adopt PaddleOCRv2 (Du et al., 2021) as a representative, a lightweight SOTA OCR system. Similar to what we've done for object detection information, we extract the textual content and corresponding central coordinates from OCR results, process them into text form, and then prepend an instruction sentence to obtain the final sentence, e.g., "Here are the central coordinates of certain texts in this image: 'Birthday'[0.41, 0.85], 'YEARS'[0.11, 0.34]."

**Examples.** Specific examples with images are provided in Appendix A.2. In Appendix B.1, we conduct statistical analyses on the length of processed texts, showing that this simple-to-implement constructing method effectively expresses useful information as well as compress the length.

### 3.2 STUDIED MODEL ARCHITECTURE

Specifically, Figure 2 illustrates the overall architecture of the studied MLLM in different training strategies, taking LLaVA-1.5 as an example. [2] Firstly, the CLIP-ViT-L-336px (Radford et al., 2021) is used to extract image-level features and a two-layer MLP is employed to align these features with text. Subsequently, we separately use DINO and PaddleOCRv2 for object detection and OCR. The results are then converted into sentences using the aforementioned methods and transformed into text features using the embedding layers of the backbone LLM. Next, we concatenate the image-level features and the detection features and input them into the backbone LLM. As a result, the MLLM can simultaneously obtain both the overall image information and the fine-grained image details during training and inference.

### 3.3 STUDIED INFUSION STRATEGIES

We systematically design three training strategies for the infusion of textual detection information, using LLaVA-1.5 as a representative. We provide more implementation details in Appendix B.

**Training-free Infusion (TFI).** For the first strategy, we directly feed the textual detection information into the MLLM without any additional training. As shown in Figure 2(a), we use the same model structure and parameter as the studied MLLM, with the only distinction being the supplementary input of the textual detection information.

**Retraining Based Infusion (RBI).** For the second strategy, we train the model from scratch using the studied MLLM's training pipeline. As shown in Figure 2(b), we first initialize the MLP module

---

[2]On the main page, we mainly focus on LLaVA-1.5 due to its architectural alignment with many leading MLLMs, making it a representative choice. For more detailed discussions and empirical evidence, please refer to Appendix D.1, where we also present findings from experiments on another MLLM, Qwen-VL, which yield similar trends to corroborate our conclusions.

and pre-train it with the studied MLLM's original pre-training dataset. Subsequently, we introduce LoRA (Hu et al., 2021) modules into the backbone LLM, Vicuna-1.5 (Chiang et al., 2023). After that, we train the LoRA modules and the MLP module during the instruction tuning process with the studied MLLM's original instruction-following dataset, whose details are provided in Appendix B.2. Throughout the entire training process, we continuously infuse the textual detection information.

**Fine-tuning Based Infusion (FTBI).** For the third strategy, we conduct fine-tuning on a well-trained MLLM. As shown in Figure 2(c), we freeze the weights of both the MLP module and the backbone LLM of the pre-trained MLLM. Following this, we introduce LoRA modules to the LLM and train the LoRA modules for a single epoch with the studied MLLM's original instruction-following dataset, concurrently infusing the textual detection information.

### 3.4 QUANTITATIVE EVALUATION SETTINGS

We employ 10 widely recognized benchmarks to evaluate different MLLM capabilities: VQAv2 (Goyal et al., 2017), GQA (Hudson & Manning, 2019), and MME (Fu et al., 2023) measure **comprehensive VQA capabilities**; MMBench (Liu et al., 2023d) and SEED-Bench (Li et al., 2023a) evaluate **perceptual and reasoning abilities**; TextVQA (Singh et al., 2019) assesses **text recognition abilities**; MM-Vet (Yu et al., 2023) evaluates **abilities for managing complex task with fine-grained image details**; and POPE (Li et al., 2023e) measures **fine-grained object localization abilities**. It's noteworthy that we evaluate the models using a subset of GQA benchmark, denoted as **GQA∗**, which retains unambiguous questions. Detailed information of the GQA∗ is provided in Appendix E.1. For a more comprehensive and convenient comparison, we compute the average percentage improvement of the models, trained with different strategies, over the original models across the 10 benchmarks, denoted as $\Delta$.

Benchmark names are abbreviated due to space limits: $VQA^{v2}$ as VQA-v2, $VQA^T$ as TextVQA, MMB as MMBench, $MMB^{CN}$ as MMBench-Chinese, SEED as SEED-Bench, $MME^P$ as MME-Perception, and $MME^C$ as MME-Cognition.

## 4 MAIN RESULTS AND ANALYSIS

### 4.1 OVERVIEW AND ORGANIZATION

In this section, we systematically evaluate the performance improvements of the enhanced MLLMs over the original models under various training strategies. We find that the FTBI training strategy yields the best results. As shown in Table 1, on 10 well-recognized MLLM benchmarks, FTBI-7B and FTBI-13B exhibits a 3.99% and 3.30% improvement compared to LLaVA-1.5-7B and LLaVA-1.5-13B respectively. Besides, FTBI-13B outperforms TFI-13B by 6.71%.

We will delve into the progressive exploration of the studied training strategies (TFI in Section 4.2, RBI in Section 4.3, and FTBI in Section 4.4). Additionally, in Section 4.5, we will test the substitution of the deployed object detection model and explore whether the fine-tuned MLLM can retain its training effects after the replacement. Moreover, in Appendix D, we will provide further experimental analysis with a new MLLM, Qwen-VL, and a new detector, YOLOv8.

### 4.2 LESSON 1: THE ORIGINAL MLLM STRUGGLE WITH COMPREHENDING TEXTUAL DETECTION INFORMATION

Initially, we input the textual detection information directly into the original LLaVA-1.5, aiming to observe whether it can comprehend and utilize this specially formatted information. We call this training strategy "Training-free Infusion" (TFI) as introduced in Section 3.3.

**Performance Improvement on OD/OCR Tasks.** The results are presented in Table 1. We can see that *TFI-7B exhibits partial enhancement in some benchmarks, while TFI-13B shows a discernible decline.* Both models show significant improvement on the POPE benchmark, which evaluates object hallucination, indicating that the infused object detection information works well. Besides, as shown in Appendix E.2, they exhibit robust performance on the MME-Cognition benchmark, which

Table 1: Comparison between "Training-free Infusion"(TFI) models, "Retraining Based Infusion" (RBI) models, "Fine-tuning Based Infusion" (FTBI) models, and the original LLaVA-1.5 on 10 benchmarks. $\Delta$ represents the average percentage improvement relative to the original models. **Bold** and underlined results indicate the best and second-best performance respectively. MME represents the summation of $\text{MME}^P$ and $\text{MME}^C$, with detailed information in Appendix E.2.

| MLLM | $\text{VQA}^{v2}$ | GQA∗ | $\text{VQA}^T$ | POPE | MME | MMB | $\text{MMB}^{CN}$ | MM-Vet | SEED | $\Delta$ |
|---|---|---|---|---|---|---|---|---|---|---|
| LLaVA-1.5-7B | 78.5 | 79.6 | 58.2 | 85.9 | 1866.4 | 64.3 | 58.3 | 30.5 | 58.6 | - |
| **TFI-7B** | 78.5 = | 79.2 ↓ | 59.2 ↑ | **89.9** ↑ | 1898.0 ↑ | 65.0 ↑ | 57.2 ↓ | 33.7 ↑ | 60.6 ↑ | +2.30% |
| **RBI-7B** | 78.5 = | 76.6 ↓ | 60.0 ↑ | 89.3 ↑ | 1866.5 ↑ | 66.2 ↑ | 60.6 ↑ | 31.5 ↑ | 60.8 ↑ | +1.91% |
| **FTBI-7B** | 79.0 ↑ | 80.1 ↑ | 60.1 ↑ | 88.9 ↑ | 1880.5 ↑ | 67.3 ↑ | 60.2 ↑ | 35.2 ↑ | 60.8 ↑ | +3.99% |
| LLaVA-1.5-13B | 80.0 | 81.0 | 61.3 | 85.9 | 1826.7 | 67.7 | 63.6 | 35.4 | 61.6 | - |
| **TFI-13B** | 76.6 ↓ | 79.0 ↓ | 59.6 ↓ | 88.3 ↑ | 1854.6 ↑ | 65.0 ↓ | 57.5 ↓ | 31.7 ↓ | 60.7 ↓ | -3.41% |
| **RBI-13B** | 79.2 ↓ | 78.0 ↓ | 61.7 ↑ | 89.2 ↑ | 1900.9 ↑ | 69.5 ↑ | 63.2 ↓ | 35.1 ↓ | **62.5** ↑ | +0.72% |
| **FTBI-13B** | **80.3** ↑ | **81.8** ↑ | 61.8 ↑ | 88.8 ↑ | **1920.5** ↑ | **71.4** ↑ | **65.2** ↑ | 38.9 ↑ | 62.3 ↑ | +3.30% |

contains numerous questions related to text within images, suggesting that the OCR information is also demonstrating efficacy.

**Performance Degradation on Other Tasks.** However, other benchmark scores exhibit fluctuations, implying a deficiency in training-free models' utilization of textual detection information. Upon closer analysis, we believe that the infusion of textual detection information introduces extraneous content, which may become noise, thereby adversely affecting the accuracy. In other words, if the models are not trained adaptively with the specially formatted detection information, it may not be able to effectively extract useful information from it and can be misguided by noise.

### 4.3 LESSON 2: RETRAINING HAS ADVERSE EFFECTS ON COMPREHENDING ViT FEATURES

In Section 4.2, we experimentally demonstrate that the studied MLLM with a training-free strategy fails to fully comprehend and use the textual detection information we input. Nevertheless, as demonstrated by numerous studies (Zhang et al., 2024b; Cabessa et al., 2024), adapting LLMs through training with specially formatted text helps them more effectively extract useful information from it, while identifying and filtering out noise within the text. Hence, we will then explore whether the retraining strategy can improve the model's understanding of this textual detection information. For the "Retraining Based Infusion" (**RBI**) strategy, we retrain LLaVA-1.5 based on its original training pipeline, concurrently infusing the textual detection information.

**Performance Improvement Relative to the Original Model.** As shown in Table 1, *RBI models excel beyond LLaVA-1.5 across several benchmarks, particularly the 7B variant*. Notably, they outshine on comprehensive benchmarks such as MMBench and Seed-Bench, and show a 4% improvement on the POPE benchmark, which assesses object hallucination. Notable gains are also seen on MME-Cognition and TextVQA, which are related to text recognition.

**Adverse Impact of Retraining on ViT Feature Comprehension.** Nevertheless, *RBI models do not show improvement across all benchmarks*. While the 13B version of RBI shows a clear advantage over the training-free model, its improvement over the original model is still limited. Besides, the 7B version of RBI even performs similarly to the training-free model. These unexpected results may be due to the redundant information in the textual detection information, which negatively affects MLLM's ability to learn how to utilize features from ViT (the image encoder) during training.

Table 2: Performance of RBI models without detection information during inference(w/o DI).

| MLLM | $\text{VQA}^{v2}$ | GQA∗ | $\text{VQA}^T$ | POPE | $\text{MME}^P$ | $\text{MME}^C$ | MMB | $\text{MMB}^{CN}$ | MM-Vet | SEED |
|---|---|---|---|---|---|---|---|---|---|---|
| LLaVA-1.5-7B | 78.5 | 79.6 | 58.2 | 85.9 | 1510.7 | 355.7 | 64.3 | 58.3 | 30.5 | 58.6 |
| **RBI-7B w/o DI** | 76.4 ↓ | 74.8 ↓ | 56.6 ↓ | 85.5 ↓ | 1387.7 ↓ | 312.5 ↓ | 65.5 ↑ | 58.3 | 29.0 ↓ | 59.6 ↑ |
| LLaVA-1.5-13B | 80.0 | 81.0 | 61.3 | 85.9 | 1531.3 | 295.4 | 67.7 | 63.6 | 35.4 | 61.6 |
| **RBI-13B w/o DI** | 77.3 ↓ | 76.0 ↓ | 58.2 ↓ | 83.4 ↓ | 1442.6 ↓ | 310.7 ↑ | 68.5 ↑ | 61.7 ↓ | 30.6 ↓ | 61.6 |

We then evaluate the performance of RBI models with no detection information applied during inference. Upon this, their benchmark scores are only related to ViT features. As shown in Table 2, the models show a noticeable performance lag compared to LLaVA-1.5, indicating that *the retraining strategy does harm the model in learning how to use image features extracted from the image encoder.* However, it is essential to note that the real world applications encompass a substantial amount of tasks that do not require fine-grained information but rather demand image-level information. Upon these tasks, the MLLM places greater reliance on ViT features. Therefore, while facilitating the model's learning of how to utilize detection information, *it is crucial to simultaneously ensure the model preserves its capability to leverage ViT features.*

## 4.4 LESSON 3: SUITABLE FINE-TUNING ACHIEVES GOOD TRADE-OFFS BETWEEN ViT FEATURES AND TEXTUAL DETECTION INFORMATION

As indicated in Section 4.3, retraining could inevitably pose challenges for MLLMs in precisely evaluating the significance of ViT features and detection information, leading to a decline in understanding ViT features and a decrease in performance on tasks unrelated to detection. For the third training strategy, we leverage the well-trained parameters of LLaVA-1.5. Specifically, we fine-tune the pre-trained LLaVA-1.5 for an additional epoch with the textual detection information infused, aiming to observe whether the fine-tuning strategy can enhance MLLMs' ability to effectively balance between ViT features and detection information, and boost their performance on fine-grained image recognition. We call this training strategy "Fine-tuning Based Infusion", abbreviated as **FTBI**.

**Performance Improvement Relative to the Original Model, the Training-free Model, and the Retrained Model.** As shown in Table 1, *both the 7B and 13B versions of FTBI exhibit superior performance compared to LLaVA-1.5, TFI, and RBI, with the FTBI models outperform the original models by up to 3.99%, and surpass the training-free models by up to 6.71%.* Simultaneously, as indicated in Table 3, when the detection information is not infused, FTBI models show significant improvement over the RBI models and achieve performance comparable to that of LLaVA-1.5, indicating that *the fine-tuning strategy retains LLaVA-1.5's original understanding of ViT features and effectively makes good trade-offs between ViT features and the detection information.*

**Performance Improvement on All Tasks.** Upon detailed analysis on Table 1, we can find that FTBI models exhibit a visible improvement on comprehensive VQA benchmarks such as $VQA^{v2}$, GQA∗, and MME. On the benchmarks that evaluate perceptual and reasoning abilities, such as MMBench and SEED-Bench, the models' performance undergoes a noticeable improvement. Moreover, the infusion of object detection information significantly improves performance on both the POPE benchmark, which evaluates object hallucinations, and the MM-Vet benchmark, which contains questions about fine-grained image recognition. Due to the infusion of OCR information, the models also exhibit commendable performance on text-related benchmarks such as TextVQA and MME-cognition. Finally, on the overall performance measure $\Delta$, FTBI models outperform LLaVA-1.5 by 3.99% and 3.30% for the 7B and 13B versions respectively. Besides, FTBI models outperform the TFI models by 1.69% and 6.71%, indicating that fine-tuning on textual detection information is effective and allows MLLMs to better comprehend and utilize the detection information.

Table 3: If we do not infuse detection information to FTBI-7B and FTBI-13B during inference, their performance will be on par with LLaVA-1.5-7B and LLaVA-1.5-13B. "w/o DI" is an abbreviation for "without detection information."

| MLLM | $VQA^{v2}$ | GQA∗ | $VQA^T$ | POPE | $MME^P$ | $MME^C$ | MMB | $MMB^{CN}$ | MM-Vet | SEED |
|---|---|---|---|---|---|---|---|---|---|---|
| LLaVA-1.5-7B | 78.5 | 79.6 | 58.2 | 85.9 | 1510.7 | 355.7 | 64.3 | 58.3 | 30.5 | 58.6 |
| **RBI-7B w/o DI** | 76.4 | 74.8 | 56.6 | 85.5 | 1387.7 | 312.5 | 65.5 | 58.3 | 29.0 | 59.6 |
| **FTBI-7B w/o DI** | 78.0 | 78.4 | 57.1 | 86.0 | 1441.8 | 303.6 | 66.9 | 59.7 | 30.1 | 60.6 |
| LLaVA-1.5-13B | 80.0 | 81.0 | 61.3 | 85.9 | 1531.3 | 295.4 | 67.7 | 63.6 | 35.4 | 61.6 |
| **RBI-13B w/o DI** | 77.3 | 76.0 | 58.2 | 83.4 | 1442.6 | 310.7 | 68.5 | 61.7 | 30.6 | 61.6 |
| **FTBI-13B w/o DI** | 79.4 | 80.0 | 60.0 | 85.3 | 1525.7 | 320.0 | 70.8 | 64.8 | 36.0 | 61.7 |

**Fine-tuned Models Can Still Perform Well Without Infusing Detection Information.** We assess the benchmark scores of FTBI models without infusing detection information during inference,

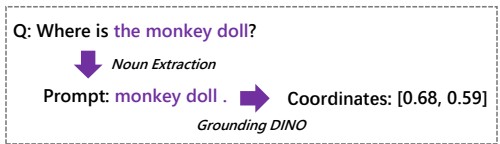

Figure 3: An example of detecting open-set targets with Grounding DINO.

Table 4: Comparison between TFI-7B and FTBI-7B employed Grounding DINO.

| | w/ Grounding DINO (box threshold 0.35) | | | | |
|---|---|---|---|---|---|
| | VQA$^{v2}$ | GQA∗ | POPE | MM-Vet | SEED |
| TFI-7B | 74.1 | 72.3 | 73.5 | 30.9 | 57.4 |
| FTBI-7B | **76.3** | **77.4** | **84.6** | **31.2** | **59.9** |

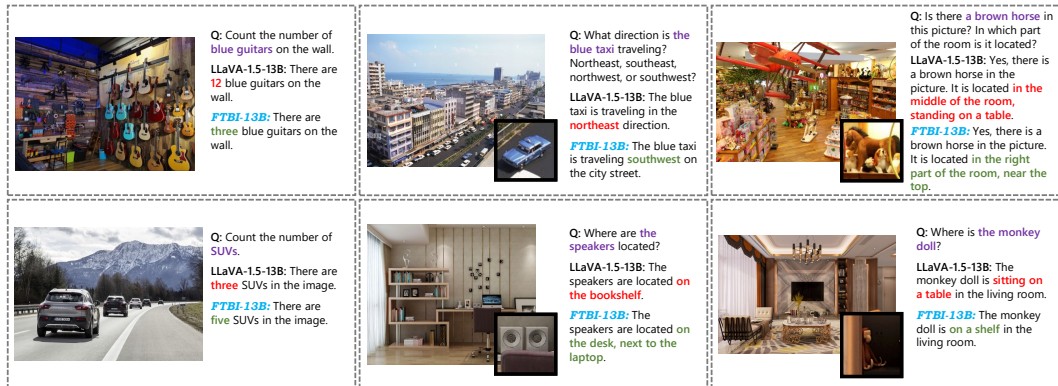

Figure 4: Examples on which LLaVA-1.5 fails while the fine-tune model (FTBI-13B) with **open-set object detection information** succeeds.

aiming to evaluate their capacities in leveraging ViT features. The findings delineated in Table 3 demonstrate that the efficacy of FTBI models without detection information aligns closely with that of LLaVA-1.5, and they outperform RBI models without detection information across all benchmarks. It means that the fine-tuning strategy effectively empowers the model to assimilate and make use of image features extracted by ViT, suggesting that it achieves a good balance between image features and detection information. Therefore, the fine-tuning strategy is superior to the training-free strategy and the retraining strategy.

### 4.5 LESSON 4: SUITABLE FINE-TUNING HELPS MLLMS BETTER UNDERSTAND SPECIALLY FORMATTED DETECTION INFORMATION

In the previous experiments, we employ DINO to extract object detection information and successfully facilitate performance improvement for the MLLM. However, it is essential to note that DINO is a closed-set object detection model, capable of detecting only 80 common object categories. Nevertheless, images may contain uncommon objects or specific entities such as certain celebrities or objects with attributive modifiers. In such scenarios, the closed-set models exhibit limitations.

Fortunately, the studied MLLM architecture is modular, and the deployed detection models are independent of the MLLM. Hence, this architecture allows for flexible replacement of the deployed detection models. In this experiment, we will substitute the closed-set detector DINO with an open-set detector to observe whether, after the replacement, the finetuned MLLMs (FTBI) can still operate effectively and acquire broader capability for detection.

**Constructing Detection Information with Grounding DINO.** In this experiment, we substitute the embedded closed-set detector DINO with an open-set object detector called Grounding DINO (Liu et al., 2023c). Grounding DINO is designed to detect objects related to user-input. With this model, the studied MLLM can locate targets by referring to the object names mentioned in questions. To achieve this, we first extract target names from the input questions and combine them to create prompts. Grounding DINO then follows the prompts to generate location information for the targets. Finally, the outputs are converted into specially formatted detection information following the method in Section 3.1. Figure 3 shows an example of this process.

**Training Effect Inherited Following Replacement of Detection Model.** In Table 4, we compare the performance of TFI-7B and FTBI-7B after replacing the detection model DINO with Grounding DINO. We use VQA$^{v2}$, GQA$*$, POPE, MM-Vet, and SEED-Bench for evaluation as they contain questions from which effective object names can be extracted. Due to the low detection accuracy of Grounding DINO, some noise is introduced, which results in lower evaluation scores for both models compared to LLaVA-1.5-7B. However, as *FTBI-7B has been fine-tuned with DINO and it can filter out some noise, the performance of FTBI-7B is superior to that of TFI-7B.* These results validate that the training effect remains after we replace the detection model.

## 5 OVERVIEW OF MORE EXPERIMENTS

We list additional experimental details and parameter settings in the appendix and conduct further experiments to validate the universality of our experimental results.

**Model Architecture Rationale.** In Appendix D.1, we discuss how does LLaVA-1.5 represents the majority of advanced MLLMs, supported by their architecture alignment. Besides, we show more empirical results on other MLLMs, Qwen-VL and LLaVA-NeXT. In Appendix D.2, we explain why DINO and PaddleOCRv2 can represent other detection models, thanks to the proposed special format. In Appendix D.3, we conduct experiments based on YOLOv5N and YOLOv11L, and investigate the impact of detector accuracy. In Appendix D.4, we remove the detection data and repeat the FTBI experiment. In Appendix D.5, we unfreeze the visual encoder and repeat the experiments. In Appendix D.6, we explore the impact of a broader object detection scope.

**Further Experiments and Analysis on the FTBI Models.** In Appendix C.1, we fine-tune LLaVA-1.5 without the infusion of detection information and discover that the exceptional performance of FTBI models is primarily ascribed to the infused detection information, rather than the additional fine-tuning. In Appendix C.2, we show the model's performance on solely leveraging object detection information or OCR information.

**Model Performance and Additional Evaluation Benchmarks.** In Appendix E.1, we elaborate on the motivations and modifications behind the GQA$*$. In Appendix E.2, we present detailed MME benchmark scores. In Appendix E.4, we evaluate our models' ability to ground specific linguistic phenomena with the VALSE benchmark. In Appendix E.3, we evaluate the models on two DocumentVQA benchmarks, DocVQA and InfographicVQA.

## 6 CONCLUSION

In this paper, we systematically conduct experiments to compare the effects of different training strategies on the infusion of textual detection information into MLLMs. After thorough investigation, we determine that fine-tuning the original MLLM for an additional epoch, along with the simultaneous infusion of textual detection information, proves to be the most effective approach compared to the training-free strategy and the retraining strategy. Moreover, we replace the detection model deployed in the studied MLLM from a close-set detector to an open-set detector and observe that the updated fine-tuned model retains the training effect and achieve better performance than the updated training-free one. This indicates that the fine-tuned model, compared to the training-free model, can better stay abreast of evolving object detection technologies and achieve sustained performance enhancements.

In a nutshell, we provide a series of progressive insights about the effective infusion of textual detection information into MLLMs. *We aim to inform researchers that when attempting to convert the outputs of vision detection models into textual information for assisting MLLMs, it can be beneficial to use a small amount of general VQA data for additional fine-tuning* (potentially using the instruction-tuning data from the MLLM itself). This approach can yield models that perform better than those not subjected to training, allowing the models to have a more comprehensive understanding and utilization of the detection information. With this work, we hope it can benefit future MLLM research and development that approaches better understanding, interpreting and engaging with fine-grained multimodal content.

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

APPENDIX

We provide more details and experiments of this work in the appendix and organize them as follows:

Appendix A, **More Demonstrative Examples**:

- Appendix A.1: we show examples on which LLaVA-1.5-13B fails while the model infused with textual detection information (FTBI-13B) succeeds.
- Appendix A.2: we show examples of images and their corresponding textual detection information, illustrating how the textual detection information is constructed.

Appendix B, **Implementation Details**:

- Appendix B.1: we conduct a statistical analysis on the length of textual detection information, showcasing the efficacy of our compression strategy.
- Appendix B.2: we introduce the instruction-following dataset of LLaVA-1.5.
- Appendix B.3: we show the different input resolutions of three branches: CLIP-ViT, DINO, and PaddleOCRv2.
- Appendix B.4: we show the thresholds we set for filtering the outputs of detection models.
- Appendix B.5: we list the training hyperparameters.
- Appendix B.6: we show the time consumption required for training models.

Appendix C, **Further Experiments and Analysis on the FTBI Model**:

- Appendix C.1: we fine-tune LLaVA-1.5 without the infusion of detection information and test the newly got models. The results indicate that the exceptional performance of FTBI models is primarily ascribed to the infused detection information, rather than the additional fine-tuning.
- Appendix C.2: we show the performance of FTBI models exclusively infusing OCR information or object detection information, affirming that they can respectively enhance the performance of MLLMs on relevant tasks.
- Appendix C.3: we assess the inference efficiency of the MLLM infused with textual detection information.

Appendix D, **Model Architecture Rationale**:

- Appendix D.1: we discuss how LLaVA-1.5 represents the majority of advanced MLLMs, and the results of LLaVA-1.5 can be extended to other MLLMs with similar structures. Additionally, we perform experiments on other MLLMs, Qwen-VL and LLaVA-NeXT, validating the versatility of our paper's experimental findings.
- Appendix D.2: we show how do DINO and PaddleOCRv2 represent other detection models in our experiments. Additionally, we perform experiments on another object detection model, YOLO-v8N, validating that the specific format we devise for processing textual detection information reduces the importance of model selection.
- Appendix D.3: we conduct experiments based on YOLOv5N and YOLOv11L, and investigate the impact of detector accuracy on MLLM performance.
- Appendix D.4: we remove the detection data from the instruction tuning dataset and repeat the FTBI experiment, aiming to investigate whether the model can still maintain good language comprehension capability.
- Appendix D.5: we introduce LoRA modules to the visual encoder and repeat the retraining and fine-tuning experiments, obtaining results consistent with the conclusions presented on the main page.
- Appendix D.6: we conduct experiments based on Co-DETR-LVIS, which is capable of detecting 1.2K object categories, to explore the impact of a broader object detection scope on MLLM performance.

- Appendix D.7: we discuss the main focus of our research, which is *"deploying detection models to assist MLLMs"*, and highlight its distinction from related works that introduce special tokens to guide MLLMs. Furthermore, we elaborate on the motivation behind focusing our research on this paradigm.

Appendix E, **Model Performance and Additional Evaluation Benchmarks**:

- Appendix E.1: we provide further details about why we modify the GQA benchmark.
- Appendix E.2: we present benchmark scores on MME-Perception and MME-Cognition.
- Appendix E.3: we evaluate our models on two well-known DocumentVQA benchmarks, DocVQA and InfographicVQA.
- Appendix E.4: we evaluate the models' ability to ground specific linguistic phenomena with the VALSE benchmark, further confirming that the fine-tuning strategy is better than the training-free strategy.

# A MORE DEMONSTRATIVE EXAMPLES

## A.1 EXAMPLES ON WHICH LLaVA-1.5-13B FAILS WHILE THE MODEL INFUSED WITH TEXTUAL DETECTION INFORMATION SUCCEEDS.

Table 5 presents examples where LLaVA-1.5 provides incorrect responses, while the FTBI-13B delivers accurate answers. The showcased examples encompass scenarios related to object counting, object detection, and text recognition.

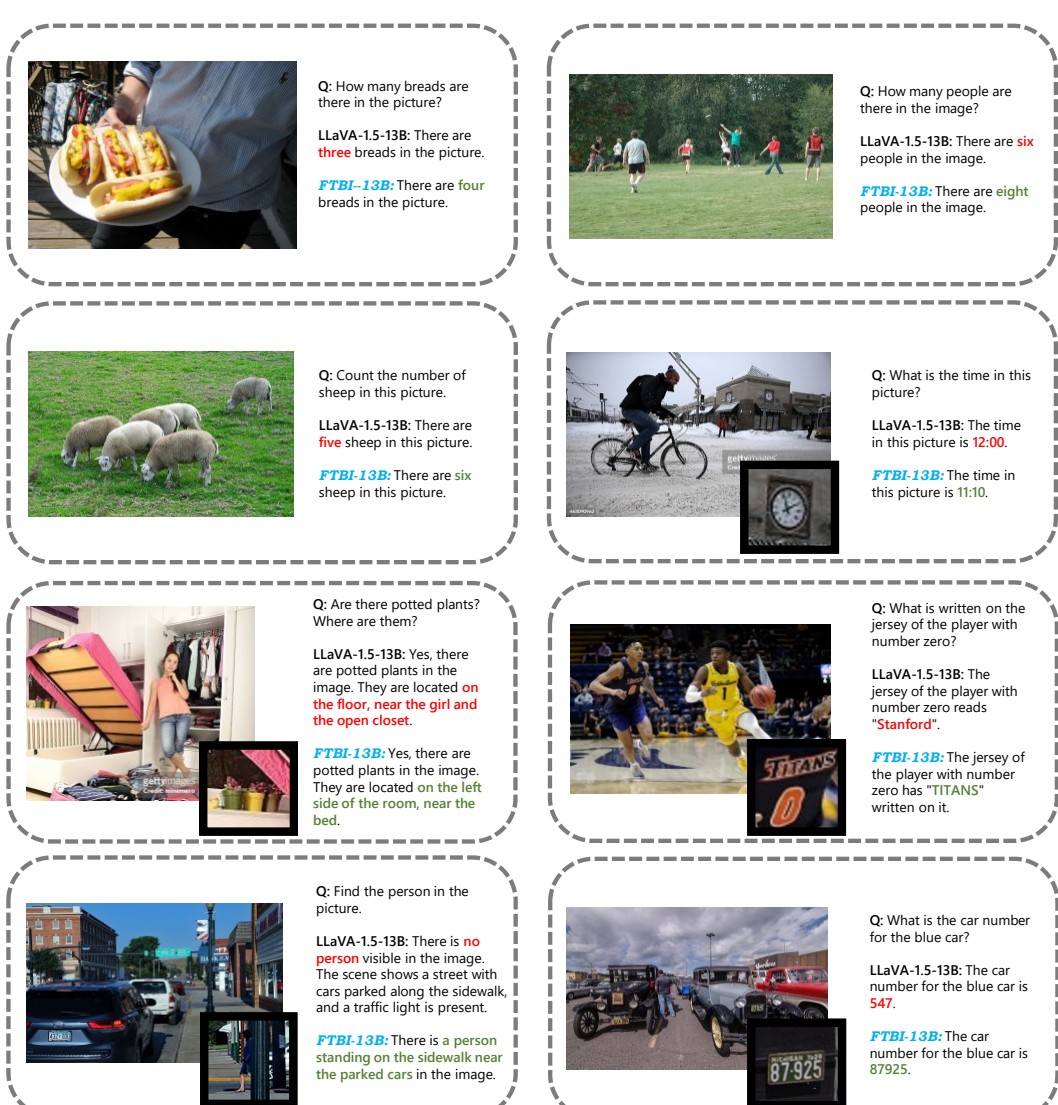

Figure 5: Examples on which LLaVA-1.5-13B fails while the model infused with textual detection information (FTBI-13B) succeeds.

## A.2 EXAMPLES OF TEXTUAL DETECTION INFORMATION

"INS: $n_1$ cls$_1$:{[pos$_1$], [pos$_2$], ...}, $n_2$ cls$_2$:{...}, ... "    "INS: text$_1$ [pos], ..."

obj$_1$(cls$_1$)   obj$_2$(cls$_1$)   obj$_3$(cls$_2$)   ...   text$_1$   ...

**Detection Models**

Figure 6: The composition of textual detection information. "INS", "obj/cls" and "pos" indicate instruction, detected object/class name, and position text respectively.

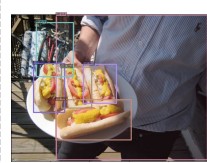

**Object Detection:** Here are the central coordinates of certain objects in this image: 1 person:{[0.61, 0.49]}, 1 bicycle:{[0.22, 0.24]}, 4 hot dog:{[0.33, 0.5], [0.19, 0.5], [0.45, 0.47], [0.42, 0.72]}.

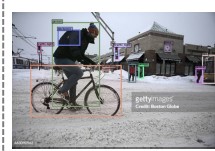

**Object Detection:** Here are the central coordinates of certain objects in this image: 4 person:{[0.31, 0.39], [0.64, 0.45], [0.59, 0.46], [0.92, 0.48]}, 1 bicycle:{[0.31, 0.58]}, 2 traffic light:{[0.52, 0.31], [0.13, 0.28]}, 1 clock:{[0.77, 0.27]}, 1 bus:{[0.97, 0.43]}, 1 train:{[0.07, 0.38]}, 1 umbrella:{[0.46, 0.42]}, 1 backpack:{[0.28, 0.19]}.
**OCR:** Here are the central coordinates of certain texts in this image: "geltyimages"[0.7, 0.64], "Credit:BostonGlobe"[0.72, 0.7], "463090962"[0.06, 0.96].

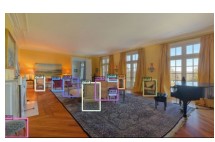

**Object Detection:** Here are the central coordinates of certain objects in this image: 8 chair:{[0.31, 0.6], [0.41, 0.67], [0.25, 0.58], [0.17, 0.57], [0.05, 0.92], [0.69, 0.6], [0.55, 0.57], [0.51, 0.53]}, 1 couch:{[0.48, 0.62]}, 2 book:{[0.51, 0.64], [0.51, 0.65]}.

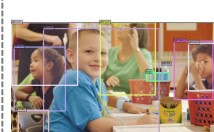

**Object Detection:** Here are the central coordinates of certain objects in this image: 6 person:{[0.43, 0.52], [0.56, 0.33], [0.16, 0.28], [0.14, 0.51], [0.88, 0.43], [0.93, 0.4]}, 1 chair:{[0.11, 0.89]}, 1 cup:{[0.7, 0.49]}, 1 bottle:{[0.74, 0.45]}.

**OCR:** Here are the central coordinates of certain texts in this image: "Crayola"[0.76, 0.8], "LARGE"[0.77, 0.88].

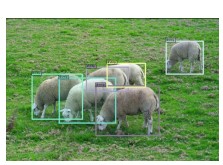

**Object Detection:** Here are the central coordinates of certain objects in this image: 6 sheep:{[0.38, 0.57], [0.24, 0.55], [0.84, 0.27], [0.57, 0.65], [0.56, 0.4], [0.47, 0.43]}.

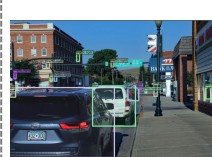

**Object Detection:** Here are the central coordinates of certain objects in this image: 11 car:{[0.59, 0.53], [0.5, 0.62], [0.25, 0.74], [0.63, 0.49], [0.97, 0.52], [0.63, 0.48], [0.57, 0.47], [0.02, 0.52], [0.6, 0.45], [0.74, 0.5], [0.56, 0.47]}, 7 traffic light:{[0.49, 0.31], [0.18, 0.32], [0.2, 0.42], [0.33, 0.27], [0.03, 0.39], [0.22, 0.42], [0.22, 0.41]}, 1 person:{[0.7, 0.5]}, 1 fire hydrant:{[0.65, 0.51]}.
**OCR:** Here are the central coordinates of certain texts in this image: "85"[0.57, 0.31], "1820"[0.54, 0.31], "1885"[0.61, 0.31], "BA"[0.69, 0.35], "K"[0.76, 0.35], "ATM"[0.77, 0.41], "432-XDU"[0.13, 0.83].

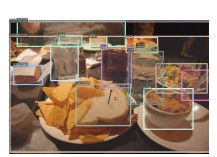

**Object Detection:** Here are the central coordinates of certain objects in this image: 5 cup:{[0.27, 0.31], [0.68, 0.43], [0.53, 0.35], [0.3, 0.15], [0.44, 0.16]}, 1 spoon:{[0.93, 0.48]}, 1 person:{[0.31, 0.1]}, 6 bowl:{[0.79, 0.66], [0.44, 0.16], [0.66, 0.18], [0.67, 0.22], [0.86, 0.43], [0.69, 0.22]}, 2 sandwich:{[0.47, 0.63], [0.09, 0.41]}, 2 dining table:{[0.5, 0.55], [0.5, 0.5]}.

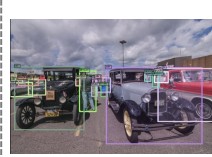

**Object Detection:** Here are the central coordinates of certain objects in this image: 5 person:{[0.38, 0.53], [0.71, 0.45], [0.67, 0.43], [0.7, 0.45], [0.33, 0.46]}, 13 car:{[0.05, 0.52], [0.7, 0.64], [0.09, 0.46], [0.46, 0.49], [0.41, 0.45], [0.13, 0.5], [0.03, 0.46], [0.74, 0.43], [0.43, 0.46], [0.19, 0.58], [0.16, 0.44], [0.04, 0.49], [0.86, 0.56]}, 2 chair:{[0.46, 0.53], [0.41, 0.53]}, 3 truck:{[0.86, 0.56], [0.09, 0.46], [0.74, 0.43]}.
**OCR:** Here are the central coordinates of certain texts in this image: "87925"[0.73, 0.62], "524"[0.19, 0.7].

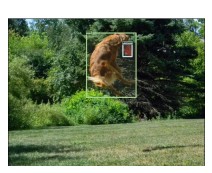

**Object Detection:** Here are the central coordinates of certain objects in this image: 1 dog:{[0.52, 0.32]}, 1 frisbee:{[0.61, 0.21]}.

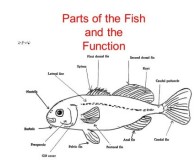

**OCR:** Here are the central coordinates of certain texts in this image: "Parts"[0.36, 0.05], "the"[0.53, 0.05], "of"[0.46, 0.05], "Fish"[0.64, 0.05], "and"[0.44, 0.14], "the"[0.54, 0.14], "Function"[0.49, 0.23], "First dorsal fin"[0.46, 0.3], "Second dorsal fin"[0.68, 0.31], "Spines"[0.39, 0.36], "Rays"[0.7, 0.38], "Lateral line"[0.26, 0.41], "Caudal peduncle"[0.8, 0.46], "Nostrils"[0.14, 0.53], "Barbels"[0.13, 0.77], "Anal fin"[0.61, 0.84], "Caudalfin"[0.77, 0.84], "Preopercle"[0.17, 0.87], "Pelvic fin"[0.36, 0.88], "Pectoralfin"[0.54, 0.88], "Gillcover"[0.23, 0.94].

Figure 7: Examples of textual detection information generated with DINO and PaddleOCRv2.

# B   IMPLEMENTATION DETAILS

## B.1   LENGTH OF TEXTUAL DETECTION INFORMATION

Since the textual descriptions of bounding box coordinates typically involve a lot of digits, their token sequences are often long. As introduced in Section 3, we devise strategies to succinctly represent the spatial information of detected objects and texts, mitigating the verbosity of bounding box descriptions. By focusing on central coordinates and consolidating objects within the same category, we maintain brevity and clarity in our model's inputs.

Table 5: The average sequence length of detection information.

|  | Average length | Average length (excluding 0) |
|---|---|---|
| Object Detection | 118.5 | 125.1 |
| OCR | 29.4 | 97.5 |

We conduct a statistical analysis on the length of detection information using samples from the instruction-following dataset of LLaVA-1.5. According to the table, the average length of object detection information is 118.5, and the average length of OCR information is 29.4. After excluding the empty sequences, the average length of object detection information rises to 125.1, while the mean length of OCR information becomes 97.5. Consequently, these numbers fall in an acceptable range and will not excessively impact the efficiency of training and inference processes.

Additionally, it is observed that approximately 0.6% of object detection information exceeds a length of 512, whereas about 0.2% of OCR information surpasses the 512 threshold. In other words, our compression strategy has effectively mitigated the occurrence of lengthy sequences.

Finally, to ensure the length of the input sequence does not exceed the maximum context window length of LLM, we exclude object detection or OCR information that exceeds a length of 1,024.

## B.2   LLAVA-1.5'S INSTRUCTION-FOLLOWING DATASET

The instruction-following dataset of LLaVA-1.5 (Liu et al., 2023a) is a combination of several datasets that relate to various tasks. Among them, the LLaVA dataset (Liu et al., 2023b) and ShareGPT dataset (Chiang et al., 2023) comprise high-quality GPT-4 conversation data. VQAv2 (Goyal et al., 2017) and GQA (Hudson & Manning, 2019) present samples that require one word or a short phrase to answer visual questions. OKVQA (Marino et al., 2019) and A-OKVQA (Schwenk et al., 2022) are VQA datasets designed to expand the knowledge base of multimodal models through the incorporation of external prior knowledge. OCRVQA (Mishra et al., 2019) is expressly tailored to enhance the text recognition capabilities of multimodal models. TextCaps (Sidorov et al., 2020) is an image captioning dataset, which presents samples in the form of concise one-sentence descriptions corresponding to images. RefCOCO (Kazemzadeh et al., 2014) and VG (Krishna et al., 2017) are object detection datasets designed to improve the object localization capabilities of multimodal models.

This dataset enables our models to better harness the additional detection information through the newly trained MLP and LoRA modules, especially with its object detection and OCR data.

Nevertheless, this dataset comprises only approximately 467K image samples, with only 116K designated for object detection and approximately 80K for text recognition, which is notably constrained. Consequently, should one seek to augment the model's proficiency in assimilating detection information effectively, the exploration of dataset expansion emerges as a viable and recommended strategy.

Regarding the pretraining dataset of LLaVA-1.5, it is pertinent to note that this dataset predominantly consists of samples tailored for image captioning, thus inherently emphasizing image-level information. However, our detection information focuses more on fine-grained details, so we opt not to incorporate this dataset in our FTBI training strategy.

## B.3 IMAGE RESOLUTION

The user-input images can be of any resolution and they are inputted into CLIP-ViT and detection modules respectively.

- For CLIP-ViT's preprocessing, input images are processed to a size of 336x336 (requiring scaling and padding to form square images).
- For DINO and Grounding DINO's preprocessing, input images can have arbitrary aspect ratios. However, we need to limit the length of the shortest side to at least 224 and the length of the longest side to be within 2048. The setting for the shortest side length is to prevent insufficient multi-scale features extracted by DINO's image encoder, ensuring an adequate number of anchor boxes. The setting for the longest side length is to reduce additional memory usage, and this value can be set arbitrarily.
- For PaddleOCRv2, we can input images of any resolution and let the model process them autonomously.

## B.4 THRESHOLD SETTING FOR DETECTION MODELS

We set certain thresholds for the detection models to reduce the acquisition of error information. Specifically, we set the threshold for DINO to 0.3 and only targets with confidence scores higher than this threshold are considered valid targets. For PaddleOCR, we set the bounding box threshold to 0.6 and only bounding boxes with confidence scores higher than this threshold are considered to contain text. For Grounding DINO, we set the bounding box threshold to 0.35 and the text threshold to 0.25, and only targets meeting the requirements of both thresholds are considered valid targets.

## B.5 TRAINING HYPERPARAMETERS

In Table 6, we show the training hyperparameters employed in our experiments. These hyperparameters are derived from Vicuna (Chiang et al., 2023) and LLaVA-1.5 (Liu et al., 2023a) and have proven to be effective. In the table, the term "Pretrain-RBI" denotes the hyperparameters used during the pre-training phase for vision-language alignment in RBI training strategy. "Finetune-RBI" refers to the hyperparameters employed for the subsequent fine-tuning phase focusing on visual instruction tuning in RBI training strategy. Additionally, "Finetune-FTBI" designates the hyperparameters used during the fine-tuning process for FTBI training strategy.

Table 6: Training hyperparameters of RBI and FTBI strategies.

| Hyperparameter | Pretrain-RBI | Finetune-RBI | Finetune-FTBI |
|---|---|---|---|
| batch size | 256 | 128 | 128 |
| MLP lr | 1e-3 | 2e-5 | - |
| lr schedule | | cosine decay | |
| lr warmup ratio | | 0.03 | |
| weight decay | | 0 | |
| optimizer | | AdamW | |
| precision | | bf16 | |
| lora rank | - | 128 | 128 |
| lora alpha | - | 256 | 256 |
| lora lr | - | 2e-4 | 2e-4 |

## B.6 TIME CONSUMPTION AND MEMORY REQUIREMENTS

As for the training cost, on four NVIDIA A100 GPUs (80GB VRAM), the time consumption in terms of the original cost and with the detection information infusion is as follows:

- For pretraining LLaVA-1.5-7B, the time increases from 6 hours to 11 hours.
- For pretraining LLaVA-1.5-13B, the time increases from 11 hours to 17 hours.

• For fine-tuning LLaVA-1.5-7B, the time increases from 16 hours to 22 hours.

• For fine-tuning LLaVA-1.5-13B, the time increases from 26 hours to 33 hours.

Regarding the memory requirements, deploying detection models results in an additional GPU memory usage of up to 4GB in each GPU compared to not deploying detection models.

# C  FURTHER EXPERIMENTS AND ANALYSIS ON THE FTBI MODELS

## C.1  FINE-TUNING ON LLAVA-1.5 WITHOUT DETECTION INFORMATION

For the FTBI training strategy, the models undergo an additional epoch of fine-tuning based on LLaVA-1.5. In the current experiment, we will train a different version of FTBI models without the infusion of detection information during training. In this way, we can investigate whether the performance improvement of the FTBI models is attributable to the supplementary detection information or to the fine-tuning of an additional epoch.

Table 7: If we finetune LLaVA-1.5 without infusing textual detection information, the performance will be inferior to the version with detection information. "-T w/o DI" stands for "training without detection information."

| MLLM | $VQA^{v2}$ | GQA* | $VQA^T$ | POPE | $MME^P$ | $MME^C$ | MMB | $MMB^{CN}$ | MM-Vet | SEED |
|---|---|---|---|---|---|---|---|---|---|---|
| FTBI-7B | 79.0 | 80.1 | 60.1 | 88.9 | 1482.7 | 397.9 | 67.3 | 60.2 | 35.2 | 60.8 |
| **FTBI-7B-T w/o DI** | 78.2 ↓ | 79.0 ↓ | 58.2 ↓ | 86.8 ↓ | 1493.0 ↑ | 345.0 ↓ | 67.3 | 60.6 ↑ | 29.8 ↓ | 60.3 ↓ |
| FTBI-13B | 80.3 | 81.8 | 61.8 | 88.8 | 1555.1 | 365.4 | 71.4 | 65.2 | 38.9 | 62.3 |
| **FTBI-13B-T w/o DI** | 79.4 ↓ | 80.7 ↓ | 60.8 ↓ | 87.1 ↓ | 1509.0 ↓ | 315.4 ↓ | 71.0 ↓ | 63.9 ↓ | 36.1 ↓ | 62.8 ↑ |

As indicated in Table 7, the performance of the models fine-tuned without infusing detection information is on par with that of LLaVA-1.5. Compared to FTBI models, these models exhibit inferior performance across almost all benchmarks. Consequently, the outstanding performance of the FTBI models is more attributed to the textual detection information we supplement, rather than that we fine-tune for an extra epoch on LLaVA-1.5.

## C.2  PERFORMANCE OF FTBI MODELS EXCLUSIVELY WITH OCR OR OBJECT DETECTION INFORMATION

Table 8: Performance of FTBI models only infused with OCR information.

| MLLM | $VQA^{v2}$ | GQA* | $VQA^T$ | POPE | $MME^P$ | $MME^C$ | MMB | $MMB^{CN}$ | MM-Vet | SEED |
|---|---|---|---|---|---|---|---|---|---|---|
| **FTBI-7B w/o DI** | 78.0 | 78.4 | 57.1 | 86.0 | 1441.8 | 303.6 | 66.9 | 59.7 | 30.1 | 60.6 |
| **FTBI-7B-OCR** | 78.3 ↑ | 78.2 | 60.3 ↑ | 86.1 | 1454.4 ↑ | 399.3 ↑ | 66.7 | 59.5 | 35.1 ↑ | 60.5 |
| **FTBI-13B w/o DI** | 79.4 | 80.0 | 60.0 | 85.3 | 1525.7 | 320.0 | 70.9 | 64.8 | 36.0 | 61.7 |
| **FTBI-13B-OCR** | 79.7 ↑ | 80.0 | 61.8 ↑ | 85.4 | 1556.9 ↑ | 367.5 ↑ | 71.1 | 65.0 | 38.0 ↑ | 61.9 |

Table 9: Performance of FTBI models only infused with object detection information.

| MLLM | $VQA^{v2}$ | GQA* | $VQA^T$ | POPE | $MME^P$ | $MME^C$ | MMB | $MMB^{CN}$ | MM-Vet | SEED |
|---|---|---|---|---|---|---|---|---|---|---|
| **FTBI-7B w/o DI** | 78.0 | 78.4 | 57.1 | 86.0 | 1441.8 | 303.6 | 66.9 | 59.7 | 30.1 | 60.6 |
| **FTBI-7B-DINO** | 79.0 ↑ | 80.1 ↑ | 57.1 | 89.0 ↑ | 1469.2 ↑ | 302.1 | 67.2 ↑ | 60.2 ↑ | 31.5 ↑ | 61.0 ↑ |
| **FTBI-13B w/o DI** | 79.4 | 80.0 | 60.0 | 85.3 | 1525.7 | 320.0 | 70.9 | 64.8 | 36.0 | 61.7 |
| **FTBI-13B-DINO** | 80.0 ↑ | 81.8 ↑ | 60.1 | 89.0 ↑ | 1529.7 ↑ | 317.9 | 71.1 ↑ | 65.0 ↑ | 37.0 ↑ | 62.3 ↑ |

As evident from Table 8 and Table 9, the infusion of object detection information boosts the scores of relevant benchmarks for object localization and object hallucination. Similarly, the infusion of OCR information improves the scores of benchmarks related to text recognition.

### C.3 INFERENCE EFFICIENCY

We assess the time consumption of the FTBI-7B model by calculating its end-to-end inference time with the GQA dataset and the TextVQA dataset. When the model relies solely on object detection information during inference, DINO accounts for 38% of the total inference time. Additionally, when OCR information is exclusively infused, PaddleOCRv2 accounts for 25% of the total inference time.

Thanks to the modularity of the studied MLLM architecture and the detection model replaceability enabled by the fine-tuning strategy, a lighter and more efficient detection model could further improve the efficiency (Wang et al., 2023b). Additionally, since the embedded detection models are mutually independent, we can let them run independently on different devices, enabling parallel inference and further accelerating inference speed.

Regarding the proposed text compression strategy (Section 3), we compare its performance with that of using the entire output from detection models (without selecting the first two values of coordinates). We find that the model with text compression achieves a 9% reduction in inference time when combined with object detection information, and a significant 58% reduction in inference time when combined with OCR information, verifying the effectiveness of the proposed text compression strategy.

## D MODEL ARCHITECTURE RATIONALE

### D.1 HOW LLAVA-1.5 REPRESENTS OTHER MLLMs?

On the main page of our paper, we exclusively select LLaVA-1.5 for experimentation, considering it representative of most advanced models. In this subsection, we will explain this choice from the following two aspects:

**(1) The representativeness of LLaVA-1.5.** We choose LLaVA-1.5 as we are in a highly dynamic field and LLaVA-1.5 is representative enough of most SOTA MLLMs. The advanced MLLMs typically consist of three main modules: an image encoder, an input projector, and a LLM backbone. LLaVA-1.5 adheres to this structure.

The process begins by encoding images into image features with an image encoder and aligning them with text features using an input projector. Most advanced MLLMs include a dedicated branch like this for processing images into analogous image token sequences. These image tokens are then combined with text tokens representing input sentences and inputted into the LLM.

Following this structure, the tokens derived from textual detection information can be directly combined with image tokens and used during MLLM's training and inference. In other words, as long as the MLLM conforms to this structure, the additional textual detection information can be processed similarly before being inputted into the LLM and serves a similar function during training and inference. Therefore, the results of experiments conducted on LLaVA-1.5 can be applied to other MLLMs with similar structures.

Furthermore, LLaVA-1.5 has proven to be highly successful, spawning numerous outstanding works. We conduct our study based on LLaVA-1.5, enabling the application of our experimental findings to the subsequent works of LLaVA-1.5.

**(2) The empirical support on Qwen-VL.** To better illustrate the versatility of our work, we also conduct experiments on another MLLM, Qwen-VL. Qwen-VL uses a cross-attention layer to compress visual features into a fixed-length sequence of 256, which differs from LLaVA-1.5's MLP. And the datasets for training are also different.

Specifically, since the instruction-following dataset of Qwen-VL-Chat is not open-sourced, *we conduct visual instruction tuning on Qwen-VL (which has not undergone visual instruction tuning) with the instruction-following dataset of LLaVA-1.5.* We compare three models: Qwen-VL-IT, Qwen-VL-IT-TFI, and Qwen-VL-IT-FTBI:

- **Qwen-VL-IT** refers to Qwen-VL undergoing regular visual instruction tuning. During the training and inference process, Qwen-VL-IT doesn't infuse textual detection information.

- **Qwen-VL-IT-TFI** follows the same training process as Qwen-VL-IT, but it infuses textual detection information during inference, corresponding to the TFI training strategy on the main page.

- **Qwen-VL-IT-FTBI** refers to fine-tuning Qwen-VL-IT while simultaneously infusing detection information during training and inference, corresponding to the FTBI training strategy on the main page.

We evaluate these models on 10 benchmarks, and the results are shown in Table 10.

Table 10: Comparison between "Qwen-VL-IT", "Qwen-VL-IT-TFI", and "Qwen-VL-IT-FTBI" on 10 well-recognized MLLM benchmark.

| | $VQA^{v2}$ | GQA∗ | $VQA^T$ | $MME^P$ | $MME^C$ | POPE | MMB | $MMB^{CN}$ | MM-Vet | SEED |
|---|---|---|---|---|---|---|---|---|---|---|
| Qwen-VL-IT | 80.8 | 82.1 | 61.7 | 1474.8 | 388.9 | 86.5 | 71.5 | 67.5 | 44.7 | 62.9 |
| Qwen-VL-IT-TFI | 80.1 ↓ | 82.5 ↑ | 61.4 ↓ | 1455.47 ↓ | 438.9 ↑ | 89.5 ↑ | 69.4 ↓ | 66.6 ↓ | 40.3 ↓ | 63.1 ↑ |
| Qwen-VL-IT-FTBI | 81.0 ↑ | 82.7 ↑ | 61.9 ↑ | 1514.3 ↑ | 417.1 ↑ | 89.5 ↑ | 72.9 ↑ | 68.6 ↑ | 46.7 ↑ | 63.1 ↑ |

Based on Table 10, it is evident that the visual grounding capability of Qwen-VL-IT-TFI has improved compared to Qwen-VL-IT, resulting in significant score increases on the POPE benchmark and the MME-Cognition benchmark. However, Qwen-VL-IT-TFI exhibits varying degrees of decline on other tasks, similar to the results of the TFI strategy on the main page.

On the other hand, Qwen-VL-IT-FTBI exhibits comprehensive improvements across all 10 benchmarks compared to Qwen-VL-IT and Qwen-VL-IT-TFI, with notable score increases in both object detection benchmarks and text recognition benchmarks. This mirrors the results of the FTBI training strategy on the main page, indicating that by infusing textual detection information during training, the model can better comprehend the detection information and consequently use it more effectively to address issues.

Table 11: If we do not infuse detection information to Qwen-VL-IT-FTBI during inference, its performance will be on par with Qwen-VL-IT. "w/o DI" is an abbreviation for "without detection information."

| | $VQA^{v2}$ | GQA∗ | $VQA^T$ | $MME^P$ | $MME^C$ | POPE | MMB | $MMB^{CN}$ | MM-Vet | SEED |
|---|---|---|---|---|---|---|---|---|---|---|
| Qwen-VL-IT | 80.8 | 82.1 | 61.7 | 1474.8 | 388.9 | 86.5 | 71.5 | 67.5 | 44.7 | 62.9 |
| Qwen-VL-IT-FTBI w/o DI | 80.6 | 81.8 | 60.9 | 1470.9 | 376.4 | 86.6 | 72.0 | 68.3 | 43.9 | 62.5 |

Additionally, as shown in Table 11, we evaluate the performance of Qwen-VL-IT-FTBI without infusing detection information during inference and find that its results are comparable to those of Qwen-VL-IT. This further supports the experimental conclusion presented in the main page: fine-tuning the original MLLM allows it to retain its ability to comprehend image features derived from the image encoder, leading to strong performance on both image-level tasks and fine-grained image recognition tasks.

**(3) The empirical support on LLaVA-NeXT.**   we conduct the FTBI experiment again using LLaVA-NeXT, aiming to investigate whether a more advanced MLLM can enhance the performance of the FTBI model. The selected base model is llama3-llava-next-8b, and the training dataset is LLaVA-NeXT's visual instruction tuning dataset. The results are presented as follows.

From Table 12, incorporating detection information improves LLaVA-NeXT's performance on benchmarks related to object detection and text recognition. Moreover, the LLaVA-NeXT version of the FTBI model demonstrates superior overall performance compared to both the original LLaVA-NeXT and the TFI model. These results align with the experimental conclusions presented on the main page.

Table 12: Comparison between "LLaVA-NeXT-8B", "LLaVA-NeXT-8B-TFI", and "LLaVA-NeXT-8B-FTBI" on 10 well-recognized MLLM benchmark.

| Model | VQA$^{v2}$ | GQA* | VQA$^T$ | MME$^P$ | MME$^C$ | POPE | MMB | MMB$^{CN}$ | MM-Vet | SEED |
|---|---|---|---|---|---|---|---|---|---|---|
| LLaVA-NeXT-8B | 82.7 | 82.8 | 65.1 | 1588.2 | 379.3 | 86.9 | 72.9 | 69.6 | 42.2 | 66.2 |
| LLaVA-NeXT-8B-TFI | 82.0 | 82.7 | 65.3 | 1525.9 | 468.9 | 90.3 | 72.0 | 70.8 | 43.8 | 65.5 |
| LLaVA-NeXT-8B-FTBI | 82.5 | 83.0 | 65.7 | 1563.9 | 445.0 | 89.4 | 74.0 | 70.3 | 44.1 | 67.0 |

In summary, we elucidate the reasons behind LLaVA-1.5's capability to serve as a representative model for many advance MLLMs. We assert that the insights drawn from experiments on LLaVA-1.5 are broadly applicable to other MLLMs with similar structure. Furthermore, we conduct additional experiments on other MLLMs, Qwen-VL and LLaVA-NeXT, thereby demonstrating the extensive validity of our research findings.

## D.2 How DINO and PaddleOCRv2 Represent Other Detecion Models?

Due to the specific textual format we designed, we can process the outputs of any object detection models and OCR models into textual detection information, as long as they can output the names of targets, the content of texts, and the corresponding coordinates of targets. (*"Here are the central coordinates of certain objects in this image: 2 people:[0.25, 0.12], [0.11, 0.43], 1 cake:[0.42, 0.32]."* or *"Here are the central coordinates of certain texts in this image: 'Birthday'[0.41, 0.85], 'YEARS'[0.11, 0.34]."* ) In other words, the selection of object detection models and OCR models is not crucial. We can choose any detection models for the experiments.

Table 13: Comparison between "LLaVA-1.5-7B", "FTBI-7B-DINO ", and "FTBI-7B-YOLOv8".

| | VQA$^{v2}$ | GQA* | VQA$^T$ | MME$^P$ | MME$^C$ | POPE | MMB | MMB$^{CN}$ | MM-Vet | SEED |
|---|---|---|---|---|---|---|---|---|---|---|
| LLaVA-1.5-7B | 78.5 | 79.6 | 58.2 | 1510.7 | 355.7 | 85.9 | 64.3 | 58.3 | 30.5 | 58.6 |
| FTBI-7B-DINO | 79.0 | 80.1 | 59.8 | 1482.7 | 397.9 | 88.9 | 67.3 | 60.2 | 35.2 | 60.8 |
| **FTBI-7B-YOLOv8** | 78.6 | 80.4 | 59.9 | 1492.1 | 400.4 | 87.2 | 68.4 | 62.5 | 34.6 | 60.2 |

To better elucidate this point, we replace DINO with another object detection model, YOLOv8, and repeat the FTBI experiments, yielding the outcomes in Table 13. According to the table, both models bring similar performance improvements to the studied MLLM, suggesting that when the functionalities and performances of detection models are similar, their impact on the MLLM's enhancement is also similar.

## D.3 Experiments on Detectors with Varying Performance

The outputs of low-performance detection models often include noise, which can adversely affect the following MLLM. To investigate the impact of detection model accuracy on the MLLM performance, we employ a low-performance detection model YOLOv5N (Jocher et al., 2023) (mAP 34.3) and a high-performance detection model YOLOv11L (mAP 53.4) (replacing only the object detection model DINO while keeping the PaddleOCR unchanged), conduct both the training-free and fine-tuning experiments again and compare the performance gains brought by them. The results are presented in Table 14.

Table 14: Experiments based on YOLOv5N and YOLOv11L.

| Model | VQA$^{v2}$ | GQA* | VQA$^T$ | MME$^P$ | MME$^C$ | POPE | MMB | MMB$^{CN}$ | MM-Vet | SEED |
|---|---|---|---|---|---|---|---|---|---|---|
| LLaVA-1.5-7B | 78.5 | 79.6 | 58.2 | **1510.7** | 355.7 | 85.9 | 64.3 | 58.3 | 30.5 | 58.6 |
| **LLaVA-1.5-7B-YOLOv5N-TFI** | 78.3 | 79.3 | 59.0 | 1459.9 | 382.9 | 86.3 | 64.2 | 56.3 | 32.2 | 59.9 |
| **LLaVA-1.5-7B-YOLOv5N-FTBI** | 78.6 | 79.9 | 60.0 | 1492.7 | 402.1 | 87.1 | 68.9 | 62.5 | 33.5 | 60.4 |
| **LLaVA-1.5-7B-YOLOv11L-TFI** | 78.5 | 79.5 | 59.0 | 1490.6 | 364.6 | 87.9 | 64.7 | 56.5 | 33.8 | 60.3 |
| **LLaVA-1.5-7B-YOLOv11L-FTBI** | **79.0** | **80.0** | **60.2** | 1497.5 | **405.4** | **88.9** | **70.3** | **62.9** | 34.6 | **60.6** |

The results are presented in the table, from which the following conclusions can be drawn:

- Under the training-free strategy, YOLOv5N introduces noise to LLaVA-1.5-7B, resulting in performance degradation. In contrast, YOLOv11L, due to its superior performance, introduces minimal noise, thereby causing negligible negative impact.

- For object detection-related tasks (POPE & MM-Vet), both YOLOv5N and YOLOv11L contribute to performance improvements under the training-free strategy. However, the improvement achieved by YOLOv5N is evidently smaller than that of YOLOv11L, which can be attributed to the disparity in their detection capabilities. This highlights the training-free strategy's limited adaptability to low-performance detection models.

- Furthermore, after fine-tuning, both two versions of the MLLM achieve comprehensive performance improvements, surpassing the original LLaVA-1.5-7B. The results align with the conclusions on the main page, demonstrating that the fine-tuning strategy enables the MLLM to better differentiate between noise and useful information and more effectively interpret specially designed detection information, leading to performance enhancement.

These results indicate that the fine-tuning strategy is more robust and better able to handle the erroneous information introduced by low-performance detection models compared to the training-free strategy.

## D.4 Model Fine-Tuning Without the Use of Detection Data

On the main page, the fine-tuning dataset we used includes object detection data. In this subsection, we will explore fine-tuning the MLLM using data unrelated to detection tasks and examine whether the FTBI model can still retain its good language understanding capabilities.

Regarding the new fine-tuning dataset, we remove samples related to "coordinate" questions (object detection samples) and eliminate all text recognition samples from the original LLaVA fine-tuning dataset. Consequently, the number of samples decreases from 665K to 450K. The experimental results are presented in the table below, and the corresponding model name is "LLaVA-1.5-7B-FTBI-FNDI".

Table 15: Results of fine-Tuning the model without using detection data.

| Model | $VQA^{v2}$ | GQA* | $VQA^T$ | $MME^P$ | $MME^C$ | POPE | MMB | $MMB^{CN}$ | MM-Vet | SEED |
|---|---|---|---|---|---|---|---|---|---|---|
| LLaVA-1.5-7B | 78.5 | 79.6 | 58.2 | 1510.7 | 355.7 | 85.9 | 64.3 | 58.3 | 30.5 | 58.6 |
| LLaVA-1.5-7B-TFI | 78.5 | 79.2 | 59.2 | 1497.0 | 401.0 | 89.9 | 65.0 | 57.2 | 33.7 | 60.6 |
| **LLaVA-1.5-7B-FTBI-FNDI** | **79.1** | 79.8 | 59.5 | **1518.0** | **410.4** | 88.8 | **68.4** | **60.3** | 33.9 | **61.1** |
| LLaVA-1.5-7B-FTBI | 79.0 | **80.1** | **60.1** | 1482.7 | 397.9 | 88.9 | 67.3 | 60.2 | **35.2** | 60.8 |

From Table 15, it is evident that even without fine-tuning on detection-related data, the FTBI model still demonstrates strong performance, significantly surpassing the original model and the training-free model. Moreover, its results are only slightly below the version fine-tuned with detection data. These results indicate that, even without fine-tuning on tasks related to detection, the fine-tuned model is still capable of maintaining a broad range of language understanding abilities.

## D.5 Model Fine-Tuning With an Unfrozen Visual Encoder

On the main page, we do not train the visual encoder because the baseline we use, LLaVA-1.5-7B, also keeps the visual encoder frozen during training. In this subsection, we unfreeze the visual encoder and repeat both the retraining and fine-tuning processes for exploration. The results are presented as follows, where "TVE" denotes training with the visual encoder unfrozen.

As shown in Table 16, even with the visual encoder being trained, the performance of the training-free, retraining, and fine-tuning strategies aligns with the patterns summarized on the main page. Specifically, the RBI model outperforms the training-free model, while the FTBI model further surpasses the RBI model. Moreover, the fine-tuned model achieves the best performance in 9 out of 10 benchmarks while training with the visual encoder unfrozen.

Table 16: Results of training with the visual encoder unfrozen.

| Model | $VQA^{v2}$ | GQA* | $VQA^T$ | $MME^P$ | $MME^C$ | POPE | MMB | $MMB^{CN}$ | MM-Vet | SEED |
|---|---|---|---|---|---|---|---|---|---|---|
| LLaVA-1.5-7B | 78.5 | 79.6 | 58.2 | 1510.7 | 355.7 | 85.9 | 64.3 | 58.3 | 30.5 | 58.6 |
| LLaVA-1.5-7B-TFI | 78.5 | 79.2 | 59.2 | 1497.0 | 401.0 | 89.9 | 65.0 | 57.2 | 33.7 | 60.6 |
| **LLaVA-1.5-7B-RBI-TVE** | 78.2 | 76.1 | 59.3 | 1466.5 | 396.4 | 89.1 | 67.2 | 60.4 | 34.0 | 60.5 |
| **LLaVA-1.5-7B-FTBI-TVE** | **79** | **79.7** | **60.4** | **1556.9** | **412.1** | 89.3 | **68.9** | **61.2** | **34.6** | **60.8** |

Table 17: Results of training with the visual encoder unfrozen (without detection information being input during inference).

| Model | $VQA^{v2}$ | GQA* | $VQA^T$ | $MME^P$ | $MME^C$ | POPE | MMB | $MMB^{CN}$ | MM-Vet | SEED |
|---|---|---|---|---|---|---|---|---|---|---|
| LLaVA-1.5-7B | 78.5 | 79.6 | 58.2 | 1510.7 | 355.7 | 85.9 | 64.3 | 58.3 | 30.5 | 58.6 |
| **LLaVA-1.5-7B-RBI-TVE w/o DI** | 76.4 | 75.4 | 56.1 | 1480.7 | 289.3 | 83.1 | 66.3 | 59.5 | 30.1 | 59.6 |
| **LLaVA-1.5-7B-FTBI-TVE w/o DI** | 78.1 | 78.9 | 57.7 | 1499.6 | 318.6 | 85.5 | 66.8 | 60.1 | 30.8 | 60.5 |

Furthermore, Table 17 presents the performance of RBI and FTBI models when the detection information is not dynamically input during inference. It demonstrates that, under the condition where the visual encoder is unfrozen, the fine-tuned model still maintains comparable performance to the original LLaVA-1.5-7B, while the RBI model performs worse than the original model. This indicates that the fine-tuning strategy better balances the contributions of the visual encoder's outputs and the detection information, thereby facilitating a more effective understanding of detection cues. These findings are consistent with the conclusions presented in our paper.

## D.6 EXPERIMENTS ON A DETECTOR WITH BROADER DETECTION RANGES

On the main page, the object detection model we use, DINO, is limited to detecting 80 object categories, as it is trained on the MS-COCO (Lin et al., 2014) dataset. In this subsection, we explore whether using an object detection model with a broader detection range could further improve the performance of the FTBI model. To this end, we select Co-DETR-LVIS (Zong et al., 2023), which is trained on the LVIS (Gupta et al., 2019) dataset and can detect 1,203 object categories. We conduct both training-free and fine-tuning experiments using Co-DETR-LVIS, and the results are as follows:

Table 18: Experimental results based on Co-DETR-LVIS.

| Model | $VQA^{v2}$ | GQA* | $VQA^T$ | $MME^P$ | $MME^C$ | POPE | MMB | $MMB^{CN}$ | MM-Vet | SEED |
|---|---|---|---|---|---|---|---|---|---|---|
| LLaVA-1.5-7B | 78.5 | 79.6 | 58.2 | **1510.7** | 355.7 | 85.9 | 64.3 | 58.3 | 30.5 | 58.6 |
| LLaVA-1.5-7B-DINO-TFI | 78.5 | 79.2 | 59.2 | 1497.0 | **401.0** | **89.9** | 65.0 | 57.2 | 33.7 | 60.6 |
| LLaVA-1.5-7B-DINO-FTBI | **79.0** | 80.1 | 60.1 | 1482.7 | 397.9 | 88.9 | **67.3** | 60.2 | 35.2 | **60.8** |
| **LLaVA-1.5-7B-CoDETR-LVIS-TFI** | 77.7 | 76.9 | 58.5 | 1465.4 | 386.8 | 87.4 | 65.7 | 57.3 | 33.9 | 60.1 |
| **LLaVA-1.5-7B-CoDETR-LVIS-FTBI** | 78.7 | 79.5 | 59.7 | 1469.1 | 387.1 | 88.4 | 66.6 | 60.1 | **35.6** | 60.7 |

We can derive the following points from the table:

- Under the training-free condition, the TFI model based on Co-DETR-LVIS performs worse than the DINO-based TFI model across almost all benchmarks. After analysis, we believe that this is because Co-DETR-LVIS introduces more noise compared to DINO, as it detects a significant number of redundant objects.

- After fine-tuning, the MLLM gains the ability to mitigate the noise introduced by Co-DETR-LVIS. Consequently, the FTBI model based on Co-DETR-LVIS achieves comprehensive performance improvements over its TFI counterpart. This observation is consistent with the conclusions presented in our paper.

- Furthermore, when comparing the FTBI model based on Co-DETR-LVIS with the FTBI model based on DINO, it is evident that the Co-DETR-LVIS-based model performs worse, exhibiting inferior results across all ten benchmarks.

In summary, detection models with a wider range of object categories do not necessarily improve the performance of the FTBI models. We think this is because many of the objects they detect are redundant and may instead introduce noise, leading to a decrease in performance scores.

### D.7 FURTHER DISCUSSION ON RELATED WORKS

**(1) Why we conduct comparative experiments around adaptive training based on "deploying detection models to assist MLLMs"?** Deploying independent detection models (or models for other downstream tasks) to generate auxiliary text for MLLMs is both straightforward and effective. By simply incorporating external text descriptions into the MLLMs, it significantly improves their performance. Moreover, the deployed models are interchangeable, allowing for convenient updates and the replacement with higher-performing models, thereby enhancing the overall performance of the framework. Given its numerous advantages, an increasing number of researchers are investigating this paradigm and working based on it.

Nevertheless, many researchers tend to adopt training-free strategies. The impact of adaptive training, however, remains an important area of investigation. Therefore, we conduct systematic experiments based on the training-free and adaptive training strategies in this paradigm, as there has not been a comprehensive comparison between them.

**(2) Distinctions from approaches involving the introduction of special tokens.** In the academic community, there is a paradigm also focusing on detection information, which involves introducing special tokens to explicitly infuse detection information in both input and output, guiding MLLMs to leverage this information. Typical methods include MiniGPT4-v2 (Chen et al., 2023b), VisionLLM (Wang et al., 2024b), and Shikra (Chen et al., 2023c).

Nevertheless, this paradigm differs significantly from the paradigm we focus on.

- First, the method of deploying detection models allows MLLMs to receive real-time detection information during both training and inference. This type of detection information encompasses the locations of all detectable objects in the image, containing rich details about the image. In contrast, the special token method, which does not deploy detection models, requires manual input of detection information at the input stage. Such detection information is typically limited to a single object or a small number of objects, serving primarily as task guidance. Thus, the role of detection information differs between these approaches: in the former, it assists MLLMs for downstream tasks by providing useful detection details, while in the latter, it usually serves only as a signal, indicating that the task involves detecting specific targets.

- Furthermore, the detection information introduced by MiniGPT4-v2 and VisionLLM is completely accurate, as it is derived from datasets. In contrast, deployed detection models may occasionally produce errors, introducing noise that affects the training-free model. This noise, however, also trains the MLLMs' ability to denoise.

Therefore, the focus of our paper is fundamentally different from them. The training strategies for deploying detection models to assist MLLMs have not been as extensively explored as methods involving the special tokens. Our systematic study on this topic represents a new departure.

**(3) Our study is a pioneering work, offering inspiration for further research.** Our research investigates whether adaptive training can help MLLMs better identify noise in real-time detection information and more effectively leverage the outputs of additional detection models to enhance VQA performance. To the best of our knowledge, no previous work has systematically explored the impact of adaptive training on deploying detection models to assist MLLMs. To draw inspiration from it, we conduct a series of systematic experiments in this direction.

Our findings demonstrate that the adaptive training strategy indeed outperforms the training-free strategy. Additionally, we confirm that fine-tuning with only a small amount of high-quality VQA data can also lead to improved performance, and the performance gain is still preserved even after replacing the detection models. As a pioneering study in this area, we have uncovered many valuable insights, and we hope our findings can provide insights for researchers in the relevant field.

# E    MODEL PERFORMANCE AND ADDITIONAL EVALUATION BENCHMARKS

## E.1    MODIFICATION ON THE GQA BENCHMARK

In the original GQA benchmark, **a response is considered correct only when it precisely matches the reference answer.** However, due to the presence of numerous synonyms in the noun vocabulary, as well as variations in noun plurality, such evaluation criteria result in the omission of many correct responses. For example, if our model provides the response "ramp" instead of the expected answer "pavement", or answers the question "what is the airplane flying above?" with "beach" instead of the expected answer "ocean", it could lead to "inaccuracies". Nonetheless, the model does not make mistakes.

Thus, we make modifications to the GQA benchmark. We select only a subset of the evaluation dataset, including samples that only require yes or no answers, as well as those involving choices (questions containing "or"). For these samples, the answer can be chosen from a limited set of options, eliminating the possibility of models providing correct but non-matching answers, which leads to more accurate evaluation outcomes. After filtering, the remaining number of samples is 5,677, approximately half of the original evaluation dataset. We name the modified evaluation benchmark as GQA∗.

## E.2    MME BENCHMARK IN TABLE 1

Table 19: Performance of TFI models, RBI models, and FTBI models on the MME benchmark.

| MLLM | MME-Perception | MME-Cognition |
|---|---|---|
| LLaVA-1.5-7B | 1510.7 | 355.7 |
| TFI-7B | 1497.0 | 401.0 |
| **RBI-7B** | 1454.5 | **412.0** |
| **FTBI-7B** | 1482.7 | 397.9 |
| LLaVA-1.5-13B | 1531.3 | 295.4 |
| TFI-7B | 1453.6 | 401.0 |
| **RBI-13B** | 1491.2 | 409.6 |
| **FTBI-13B** | **1555.1** | 365.4 |

In Table 19, we present benchmark scores for TFI models, RBI models, and FTBI models on MME-Perception and MME-Cognition. According to the table, it reveals a significant enhancement in scores for both models on MME-Cognition. This notable enhancement can be ascribed to the infusion of supplementary OCR information, addressing a multitude of questions within MME-Cognition that pertain to textual content embedded within images.

Furthermore, concerning the MME-Perception benchmark, our models exhibit some fluctuations in scores. Nonetheless, it is noteworthy that the scores for FTBI models surpass those for TFI models and RBI models, which underscores that the fine-tuning approach better preserves the original image understanding capabilities of MLLMs.

## E.3    PERFORMANCE ON DOCUMENTVQA BENCHMARKS

In this subsection, we evaluate our models on two well-known DocumentVQA benchmarks, DocVQA (Mathew et al., 2021) and InfographicVQA(Mathew et al., 2022). These benchmarks are specifically designed for visual question answering tasks where questions are answered using text

within the document images. Their datasets provide OCR transcriptions and ground truth answers, enabling the evaluation of models in interpreting and extracting information from documents.

The results are presented in the two tables below. The first table compares the performance of the TFI, RBI, and FTBI models on the DocVQA and InfographicVQA benchmarks. The second table compares the performance of the RBI and FTBI models on the same benchmarks without incorporating detection information during inference.

Table 20: Performance of the TFI, RBI, and FTBI models on DocVQA and InfographicVQA.

| Model | DocVQA | InfographicVQA |
|---|---|---|
| LLaVA-1.5-7B | 19.4 | 18.8 |
| **LLaVA-1.5-7B-TFI** | 35.3 | 21.0 |
| **LLaVA-1.5-7B-RBI** | 35.7 | 20.9 |
| **LLaVA-1.5-7B-FTBI** | 35.9 | 21.3 |
| | | |
| LLaVA-1.5-13B | 20.6 | 20.7 |
| **LLaVA-1.5-13B-TFI** | 35.5 | 22.1 |
| **LLaVA-1.5-13B-RBI** | 37.9 | 23.3 |
| **LLaVA-1.5-13B-FTBI** | 38.5 | 24.2 |

Table 21: Performance of the TFI, RBI, and FTBI models on DocVQA and InfographicVQA (without detection information being input during inference).

| Model | DocVQA | InfographicVQA |
|---|---|---|
| LLaVA-1.5-7B | 19.4 | 18.8 |
| **LLaVA-1.5-7B-RBI w/o DI** | 17.3 | 17.8 |
| **LLaVA-1.5-7B-FTBI w/o DI** | 19.4 | 18.7 |
| | | |
| LLaVA-1.5-13B | 20.6 | 20.7 |
| **LLaVA-1.5-13B-RBI w/o DI** | 18.6 | 20.1 |
| **LLaVA-1.5-13B-FTBI w/o DI** | 20.6 | 20.9 |

As shown in Table 20, the deployment of detection models, particularly the OCR model, leads to a significant score improvement on DocVQA. Furthermore, models with adaptive training noticeably outperform training-free models . Specifically, the FTBI models surpass the RBI models, which in turn outperforms the TFI models. This suggests that the adaptive training enables MLLMs to better leverage the input detection information, resulting in improved performance.

Table 21 presents a comparison between the RBI models and the FTBI models in the absence of infused detection information. As shown, the performance of the RBI models is significantly inferior to that of the FTBI models. While the FTBI models, without detection information, perform similarly to the original LLaVA-1.5. This demonstrates that the fine-tuning strategy allows MLLMs to effectively balance the weights between the image encoder output and textual detection information, thereby preserving the comprehensive VQA capabilities. These results are consistent with the findings on the main page.

### E.4 PERFORMANCE ON THE VALSE BENCHMARK

VALSE (Parcalabescu et al., 2022) (Vision And Language Structured Evaluation) is a zero-shot benchmark designed to test the visual-linguistic grounding capabilities of general-purpose vision-language models on specific linguistic phenomena. It assesses many capabilities of MLLMs, including six aspects: existence, plurality, counting, spatial relations, actions, and entity co-reference. In this subsection, we will evaluate the performance of LLaVA-1.5, TFI-7B, and FTBI-7B on the VALSE benchmark and compare their results. This analysis will further validate our conclusion on the main page: the fine-tuning strategy enables MLLMs to better understand the input texual detection information compared to the training-free approach.

Table 22: Comparison between LLaVA-1.5-7B, TFI-7B, and FTBI-7B on the VALSE benchmark.

| | | Existence | Plurality | Counting_hard | Counting_small |
|---|---|---|---|---|---|
| | LLaVA-1.5-7B | 69.9 | 13.4 | 35.9 | 35.6 |
| $acc_r$ | TFI-7B | **74.1** | 9.3 | 38.0 | 40.9 |
| | **FTBI-7B** | 70.5 | **17.6** | **46.1** | **51.6** |
| | LLaVA-1.5-7B | 84.0 | 56.2 | 64.6 | 66.9 |
| $acc$ | TFI-7B | **85.9** | 54.6 | 66.1 | 68.7 |
| | **FTBI-7B** | 84.1 | **58.1** | **71.1** | **74.4** |
| | LLaVA-1.5-7B | 73.7 | 16.0 | 52.1 | 45.7 |
| $min(p_c, p_f)$ | TFI-7B | **77.6** | 11.5 | 57.3 | 51.3 |
| | **FTBI-7B** | 71.5 | **22.1** | **69.5** | **65.3** |
| | | Counting_adversarial | Relations | Action Replacement | Actant Swap |
| | LLaVA-1.5-7B | 25.2 | 4.7 | 34.3 | 10.3 |
| $acc_r$ | TFI-7B | 24.0 | 2.4 | 29.9 | 11.2 |
| | **FTBI-7B** | **36.3** | **8.2** | **37.4** | **19.2** |
| | LLaVA-1.5-7B | 55.6 | 52.0 | 66.4 | 53.1 |
| $acc$ | TFI-7B | 55.1 | 50.9 | 64.4 | 55.0 |
| | **FTBI-7B** | **64.8** | **53.4** | **67.6** | **57.3** |
| | LLaVA-1.5-7B | 39.8 | 7.5 | 43.8 | 16.7 |
| $min(p_c, p_f)$ | TFI-7B | 41.2 | 4.7 | 35.7 | 16.5 |
| | **FTBI-7B** | **59.5** | **14.6** | **52.9** | **30.2** |
| | | Coreference | Coreference_hard | Foil_it | |
| | LLaVA-1.5-7B | 5.2 | 4.8 | 50.5 | |
| $acc_r$ | TFI-7B | 3.1 | 3.9 | 56.8 | |
| | **FTBI-7B** | **20.2** | **18.3** | **63.0** | |
| | LLaVA-1.5-7B | 52.3 | 52.4 | 75.1 | |
| $acc$ | TFI-7B | 51.3 | 51.4 | 78.4 | |
| | **FTBI-7B** | **58.6** | **55.8** | **81.4** | |
| | LLaVA-1.5-7B | 6.4 | 4.8 | 53.5 | |
| $min(p_c, p_f)$ | TFI-7B | 3.8 | 3.9 | 58.3 | |
| | **FTBI-7B** | **24.0** | **20.2** | **66.6** | |

In VALSE, a valid instance consists of an image, a caption, and a modified caption called a 'foil' that exemplifies a specific linguistic phenomenon. The tested model is required to distinguish between real captions and foils. VALSE employs four metrics to evaluate the model's performance: overall accuracy ($acc$) on all classes (foil and correct); precision ($p_c$) measuring how well models identify the correct examples; foil precision($p_f$) measuring how well foiled cases are identified; and pairwise ranking accuracy ($acc_r$), which measures whether the image-sentence alignment score is greater for a correct image-text pair than for its foiled pair. $acc_r$ is more permissive than $acc$ as it consider the model prediction correct if the score for a foil is lower than the score for a caption.

Due to the inability of LLaVA-1.5 and our models to directly output "cross_relationship_score" as the image-sentence alignment score like models such as LXMERT, we modify the computation of $acc_r$, $acc$, $p_c$ and $p_f$ following the approach outlined in "lxmert_valse_eval.py" (https://github.com/Heidelberg-NLP/VALSE/blob/main/lxmert_valse_eval.py) as follows:

(1) Let the model answer the following two questions and tally the number of 'yes' and 'no' responses for each question.:

- Q1: "Does this image match the sentence 'caption'? Use only 'yes' or 'no' to answer."
- Q2: "Does this image match the sentence 'foil'? Use only 'yes' or 'no' to answer."

(2) When the answer to Question 1 is "yes", increment the counters for *foil_accuracy* and *capt_fits*. When the answer to Question 2 is "no", increment the counters for *foil_detected* and *foil_accuracy*. If the answer to Question 1 is "yes" and the answer to Question 2 is "no", increment the counter for *pairwise_acc*.

(3) The final calculation formula is:

$$acc = \frac{foil\_accuracy}{count} * 50,$$

$$p_c = \frac{capt\_fits}{count} * 100,$$

$$p_f = \frac{foil\_detected}{count} * 100,$$

$$acc_r = \frac{pairwise\_acc}{count} * 100$$

The results are presented in Table 22. It can be observed that TFI-7B performs better than LLaVA-1.5-7B in some areas, while FTBI-7B outperforms LLaVA-1.5-7B in all aspects, which indicates that the models infused with textual detection information are more sensitive to foiled instances and have better capabilities in visual grounding. Moreover, FTBI-7B outperforms TFI-7B on all metrics except for the "Existence" metric, further demonstrating that fine-tuning strategies are more effective than training-free approaches in helping MLLMs understand and utilize textual detection information.

