# OpenReview forum: "From Training-Free to Adaptive: Empirical Insights into MLLMs' Understanding of Detection Information"
_ICLR.cc/2025/Conference — Submitted to ICLR 2025_

### Official Review · Reviewer_YDqj · 2024-10-31

**Soundness:** 3
**Presentation:** 3
**Contribution:** 3
**Rating:** 6
**Confidence:** 5

**Summary:**

MLLMs struggle with accurately interpreting fine-grained visual details. While vision detection models excel at this, most studies simply insert the detection information as text into the MLLM without further training (training-free).  This paper investigates whether adaptive training can improve the MLLM's understanding of this added textual detection information, leading to better performance than training-free methods.

**Strengths:**

1. Reasonable motivation. Additional vision experts can further enhance the visual capacities of MLLMs. The paper finds the adaptive training can achieve great potential for helping LLMs better comprehend the special detection input.

2. The conducted experiments and visualizations are extensive and well-organized.

3. The paper is well-written and easy to understand.

**Weaknesses:**

1. The paper lacks analysis regarding the impact of detector performance. Would a detector with significantly higher mAP lead to greater MLLM improvement?

2. Detectors trained on larger datasets with more categories, such as LVIS (1.2k categories) compared to COCO (80 categories), potentially achieve finer-grained visual understanding. Would using the LVIS-trained detector, like Co-DETR-LVIS [1], improve FTBI performance?

3. The proposed method with an open-set detector is similar to VisualCOT [2]. Both first locate the box region that is relevant to the user question and leverage the region information to help MLLM better answer the question.

4. Can FTBI further improve performance upon stronger open-source baselines like LLaVA-NeXT [3] and LLaVA-OneVision [4]?

5. There are two paradigms to incorporate detection experts into MLLMs in the community. One converts detector outputs directly into text descriptions for the MLLM (as in this paper, MoAI [5] and IVE [6]), while the other fuses detector vision backbones with CLIP features (MoVA [7] and Eagle [8]). What advantages does the former approach offer?

[1] Detrs with collaborative hybrid assignments training. ICCV 2023.

[2] Visual cot: Unleashing chain-of-thought reasoning in multi-modal language models. NeurIPS 2024.

[3] Llava-next: Improved reasoning, ocr, and world knowledge.

[4] Llava-onevision: Easy visual task transfer.

[5] Moai: Mixture of all intelligence for large language and vision models. ECCV 2024.

[6] Incorporating visual experts to resolve the information loss in multimodal large language models.

[7] Mova: Adapting mixture of vision experts to multimodal context. NeurIPS 2024.

[8] Eagle: Exploring the design space for multimodal llms with mixture of encoders.

**Questions:**

Please see the Weaknesses.

---

> ### Author Response · Authors · 2024-11-23
>
> # Responses to Reviewer YDqj [Part 1/3]
>
> ---
>
> Dear Reviewer YDqj, we sincerely thank you for your recognition of the novelty and comprehensive experiments in our work! Following your constructive suggestions, we conduct several additional experiments, which require considerable time and effort. Below, we provide detailed responses to each of your comments, with the hope that this will encourage you to lean toward the acceptance of our paper.
>
> ---
>
> ## Response Overview
>
> 1. **More Experiments**
>
> - W1:  We conduct experiments using YOLOv5N and YOLOv11L, and compare the impact of detector performance on the performance improvement of MLLMs.
> - W2:  We conduct experiments based on Co-DETR-LVIS to explore the impact of a broader object detection scope on MLLM performance.
> - W4: We conduct experiments based on LLaVA-NeXT and validate that our conclusions remain effective on a stronger MLLM.
>
> 2. **Clarification of the Reviewer's Misunderstanding**
>
> - W3: We clarify that the experiment involving an open-set detector is for investigating whether the training effects brought by the fine-tuning can be inherited, rather than proposing a new method.
> - W5:  We explain that the two paradigms of integrating detection models do not have a clear advantage over each other. Additionally, we clarify why our work focuses on the first approach.
>
> ---
>
> ## W1: The Impact of Detector Performance
>
> The selected object detection model, DINO, achieves a high mAP score of 58.5 on the COCO benchmark. It is challenging to identify another object detection model that offers significantly better performance than DINO while maintaining a comparable fast inference speed. To address your concern, **we conduct the FTBI experiments again based on two object detection models with markedly different performance levels for comparison.** They are YOLOv5N (mAP 34.3) and YOLOv11L (mAP 53.4). The new results are presented as follows:
>
>
>
> | Model                          | VQA$^{v2}$ | GQA*   | VQA$^T$  | MME$^P$    | MME$^C$   | POPE     | MMB      | MMB$^{CN}$ | MM-Vet   | SEED     |
> | ------------------------------ | ---------- | ------ | -------- | ---------- | --------- | -------- | -------- | ---------- | -------- | -------- |
> | **LLaVA-1.5-7B**               | 78.5       | 79.6   | 58.2     | **1510.7** | 355.7     | 85.9     | 64.3     | 58.3       | 30.5     | 58.6     |
> | **LLaVA-1.5-7B-YOLOv5N-TFI**   | 78.3       | 79.3   | 59.0     | 1459.9     | 382.9     | 86.3     | 64.2     | 56.3       | 32.2     | 59.9     |
> | **LLaVA-1.5-7B-YOLOv5N-FTBI**  | 78.6       | 79.9   | 60.0     | 1492.7     | 402.1     | 87.1     | 68.9     | 62.5       | 33.5     | 60.4     |
> | **LLaVA-1.5-7B-YOLOv11L-TFI**  | 78.5       | 79.5   | 59.0     | 1490.6     | 364.6     | 87.9     | 64.7     | 56.5       | 33.8     | 60.3     |
> | **LLaVA-1.5-7B-YOLOv11L-FTBI** | **79**     | **80** | **60.2** | 1497.5     | **405.4** | **88.9** | **70.3** | **62.9**   | **34.6** | **60.6** |
>
> We can summarize the following points:
>
> - Under the training-free strategy, for general VQA capabilities, YOLOv5N introduces noise to LLaVA-1.5-7B, resulting in performance degradations. In contrast, YOLOv11L, due to its superior performance, introduces minimal noise and thus has little negative impact.
> - Regarding the object detection-related capabilities, both YOLOv5N and YOLOv11L bring performance improvements under the training-free strategy. However, the improvement from YOLOv5N is noticeably smaller than that from YOLOv11L, which can be attributed to the different model performance.
> - Furthermore, after fine-tuning, both two versions of the MLLM achieve comprehensive performance improvements, surpassing the original LLaVA-1.5-7B.  Moreover, the model with YOLOv11L consistently outperforms the model with YOLOv5N across all benchmarks.
>
> Therefore, high-performing object detection models can indeed bring more performance gains to MLLMs than low-performing object detection models. Thank you for your suggestion! We have included this discussion in Appendix D.3 of our revised paper.
>
> (To be continued.)

---

> ### Author Response · Authors · 2024-11-23
>
> # Responses to Reviewer YDqj [Part 2/3]
>
> ---
> (Continued from the previous page. W1)
>
>
> ## W2: "Would using the LVIS-trained detector improve FTBI performance?"
>
> As per your suggestion, we conduct the training-free and fine-tuning experiments using **Co-DETR-LVIS**, aiming to investigate whether a closed-set detection model with a broader detection range could further enhance the performance of the FTBI model.  The new results are as follows:
>
> | Model                             | VQA$^{v2}$ | GQA*     | VQA$^T$  | MME$^P$    | MME$^C$   | POPE     | MMB      | MMB$^{CN}$ | MM-Vet   | SEED     |
> | --------------------------------- | ---------- | -------- | -------- | ---------- | --------- | -------- | -------- | ---------- | -------- | -------- |
> | LLaVA-1.5-7B                      | 78.5       | 79.6     | 58.2     | **1510.7** | 355.7     | 85.9     | 64.3     | 58.3       | 30.5     | 58.6     |
> | LLaVA-1.5-7B-DINO-TFI             | 78.5       | 79.2     | 59.2     | 1497.0     | **401.0** | **89.9** | 65.0     | 57.2       | 33.7     | 60.6     |
> | LLaVA-1.5-7B-DINO-FTBI            | **79.0**   | **80.1** | **60.1** | 1482.7     | 397.9     | 88.9     | **67.3** | **60.2**   | 35.2     | **60.8** |
> | **LLaVA-1.5-7B-CoDETR-LVIS-TFI**  | 77.7       | 76.9     | 58.5     | 1465.4     | 386.8     | 87.4     | 65.7     | 57.3       | 33.9     | 60.1     |
> | **LLaVA-1.5-7B-CoDETR-LVIS-FTBI** | 78.7       | 79.5     | 59.7     | 1469.1     | 387.1     | 88.4     | 66.6     | 60.1       | **35.6** | 60.7     |
>
>  We can derive the following points from the table:
>
> - Under the training-free condition, the TFI model based on Co-DETR-LVIS performs worse than the DINO-based TFI model across almost all benchmarks. After analysis, we believe that this is because Co-DETR-LVIS introduces more noise compared to DINO, as it detects a significant number of redundant objects.
> - After fine-tuning, the MLLM gains the ability to mitigate the noise introduced by Co-DETR-LVIS. Consequently, the FTBI model based on Co-DETR-LVIS achieves comprehensive performance improvements over its TFI counterpart. This observation is consistent with the conclusions presented in our paper.
> - Furthermore, when comparing the FTBI model based on Co-DETR-LVIS with the FTBI model based on DINO, it is evident that the Co-DETR-LVIS-based model performs significantly worse, exhibiting inferior results across all nine benchmarks.
>
> In summary, detection models capable of identifying a wider range of objects do not necessarily improve the performance of the FTBI models. We think this is because many of the objects they detect are redundant and may instead introduce noise, leading to a decrease in performance scores. We have included these additional experiments in Appendix D.6 of our revised paper.
>
>
>
> ---
>
> ## W3: "The proposed method with an open-set detector is similar to VisualCoT."
>
> The purpose of introducing the experiment related to Grounding DINO **is not to propose a new method**, but rather to verify that our fine-tuned model's effectiveness can still be maintained after replacing the detection model. Grounding DINO introduces a lot of noise, and it generates information differing from DINO's due to the differences in detection range. In light of this, our aim is to validate that the fine-tuned model can retain its denoising capabilities and understanding of specialized detection information**, after the detection model being replaced without additional training**.
>
> Here, we aim to expand on the ablated factor for our systematic study on detection information infusion, rather than proposing a new method.
>
> (To be continued.)

---

> ### Author Response · Authors · 2024-11-23
>
> # Responses to Reviewer YDqj [Part 3/3]
>
> ---
> (Continued from the previous page. W2&W3)
>
> ## W4: "Can FTBI further improve performance upon stronger open-source baselines?"
>
> Based on your suggestion, we conduct the fine-tuning experiment using LLaVA-NeXT, aiming to investigate whether a more advanced MLLM can enhance the performance of the FTBI model. The selected base model is llama3-llava-next-8b, and the training dataset is LLaVA-NeXT's visual instruction tuning dataset. The new results are presented as follows.
>
> | Model                  | VQA$^{v2}$ | GQA* | VQA$^T$ | MME$^P$ | MME$^C$ | POPE | MMB  | MMB$^{CN}$ | MM-Vet | SEED |
> | ---------------------- | ---------- | ---- | ------- | ------- | ------- | ---- | ---- | ---------- | ------ | ---- |
> | LLaVA-NeXT-8B          | 82.7       | 82.8 | 65.1    | 1588.2  | 379.3   | 86.9 | 72.9 | 69.6       | 42.2   | 66.2 |
> | **LLaVA-NeXT-8B-TFI**  | 82.0       | 82.7 | 65.3    | 1525.9  | 468.9   | 90.3 | 72.0 | 70.8       | 43.8   | 65.5 |
> | **LLaVA-NeXT-8B-FTBI** | 82.5       | 83.0 | 65.7    | 1563.9  | 445.0   | 89.4 | 74.0 | 70.3       | 44.1   | 67.0 |
>
> From the table, incorporating detection information improves LLaVA-NeXT's performance on benchmarks related to object detection and text recognition. Moreover, the LLaVA-NeXT version of the FTBI model demonstrates superior overall performance compared to both the original LLaVA-NeXT and the TFI model. These results align with the experimental conclusions presented in our paper.
>
> Thank you for your suggestion. We have included this subsection in Appendix D.1 of our revised paper.
>
>
>
> ---
>
> ## W5: The Two Paradigms to Incorporate Detection Experts
>
> We have compared these two paradigms in our related works (lines 150–187). As mentioned, the former approach is often similar to multi-agent methodologies, being plug-and-play and requiring no training. Whereas the latter often necessitates training for new adapters and may require substantial amounts of data. However, with ongoing advancements in this field, the need for training data is becoming less extensive. So it will not be a disadvantage in the future.
>
> When determining the research focus of our paper, we have conducted comparative experiments on these two approaches. Specifically, we evaluated three methods: (1) directly converting the outputs of detection models into textual description, (2) using a newly initialized MLP layer to map the output features of detection models , and (3) fusing detection features with CLIP features using a six-layer Cross Attention mechanism. Based on these methods, we incorporated the outputs of detection models into MLLMs and conducted re-training of LLaVA-1.5-7B through both pre-training and fine-tuning.  Finally, we evaluated these methods on the VQA-v2, TextVQA, MME-Cognition, and POPE benchmarks, with the results summarized as follows:
>
> |                 | VQA$^{v2}$ | VQA$^{T}$ | MME$^{C}$ | POPE​ |
> | --------------- | ---------- | --------- | --------- | ---- |
> | MLP             | 77.7       | 57.5      | 268.6     | 86.9 |
> | Cross Attention | 77.2       | 57.4      | 265.4     | 85.9 |
> | Text-based      | 78.5       | 60.0      | 412.9     | 89.3 |
>
> Under the same training conditions, the text-based approach achieves the best performance, effectively transferring the outputs of detection models to the MLLM and enhancing its capabilities. In contrast, due to the limited amount of training data, the newly initialized structures of the MLP and Cross-Attention methods cannot be adequately trained, resulting in suboptimal performance. Therefore, based on our comparisons, we believe that conducting adaptive training research centered on the text-based approach is more likely to yield significant results.
>
> Many researchers in the relevant field adopt the former approach for their studies, as it is more straightforward and delivers noticeable results. By simply incorporating external text descriptions into the MLLMs, they can significantly improve the performance of MLLMs. We conduct experiments around the former because **there has not been a comprehensive comparison between training-free and adaptive training methods** using this approach, and we hope our findings will inspire researchers who use the former approach.
>
> Besides, the references you provide are highly valuable, and we have included them in the related works section of our revised paper.
>
>
> ---
>
> ## Closing Remarks
>
> We sincerely appreciate the valuable time you have dedicated to reviewing our work, as well as the insightful feedback you provide. As per your suggestions, we invest considerable time and effort into conducting additional experiments.  We hope these experiments address your concerns and lead you to raise the score. Thank you once again for your time and effort.

---

> > ### Comment · Reviewer_YDqj · 2024-11-24
> >
> > Thanks for your response. I will increase my rating to 6.

---

> > > ### Author Response · Authors · 2024-11-26
> > >
> > > Dear Reviewer YDqj,
> > >
> > > Thank you for your attention to our response and the efforts you've dedicated to reviewing our paper! It is encouraging to note your response to raise your rating, favoring the acceptance of our work.
> > >
> > > We believe that your comments and these responses have further enhanced the quality and contribution of our work. Should you require any additional information or have further questions, we are readily available to engage more. Thank you once again for your time and helpful feedback!
> > >
> > > Best, authors

---

### Official Review · Reviewer_JV9J · 2024-11-01

**Soundness:** 2
**Presentation:** 2
**Contribution:** 1
**Rating:** 5
**Confidence:** 5

**Summary:**

Inspired by the absence of adaptive training methods to integrate textual detection information into MLLMs, the paper empirically explored the effect of fine-tuning MLLMs equipped with textual detection information. The key insights were 1) fine-tuning strategy yields better performance than training-free and retraining strategies, 2) retraining rather impairs the original image comprehension ability of MLLMs, and 3) Swapping the deployed detection model with open-set object detector further improves MLLM performance.

**Strengths:**

- The paper conducted a comprehensive set of experiments with thorough analysis for integrating textual detection information into MLLM
- The empirical observations are straightforward and the paper is written in easy-to-understand manner

**Weaknesses:**

- The key insights and the empirical observations this paper investigated may seem to reiterate the existing observations of the related papers. Specifically, [MiniGPT4-v2](https://arxiv.org/abs/2310.09478) and  [VisionLLM](https://arxiv.org/abs/2305.11175) are the pioneering works that demonstrated the positive impact of integrating object detection information into MLLMs in terms of object detection and several MLLM tasks (e.g., VQA).
- Additionally, the paper overlooks the effectiveness of training-free methods, which avoid the need for a huge amount of labor-intensive annotations required for equipping such large-scale MLLMs with object detection ability.
- The novelty of the proposed methods is significantly limited, which is a simple adoption of training modules onto training-free infusion models.
- The technical soundness of the proposed methods seems deficient. Why is the retraining strategy that trains MLLM from scratch not training visual encoders? Also, there is no justification for why the different backbone (vicuna) is used for retraining, compared to the other fine-tuning and training-free strategies.

**Questions:**

See above weaknesses

---

> ### Author Response · Authors · 2024-11-23
>
> # Responses to Reviewer JV9J [Part 1/3]
> ---
>
> Dear Reviewer JV9J, we sincerely appreciate the time and effort you have dedicated to reviewing our paper, as well as the valuable feedback and suggestions you provide! However, after carefully reading your response,  we believe you may have misunderstood certain aspects of our work. In light of this, we address your concerns one by one in our rebuttal and hope that you will consider re-evaluating our paper.
>
> ---
>
>
>
> ## Response Overview
>
> We give a quick summarization of our response and present more details later.
>
> 1. **More Experiments**
>
> - W4.1: We explain that we don't train the visual encoder because the baseline model we choose, LLaVA-1.5, does not train it either. To address the reviewer's concern, we unfreeze the visual encoder and conduct additional experiments, which yields results consistent with the conclusions of our paper.
>
> 2.  **Clarification of the Reviewer's Misunderstanding**
>
> - W1:  We explain the significant differences between our paper and related works such as MiniGPT4-v2 and VisionLLM.
> - W2: We clarify that we don't overlook the efficiency of training-free methods, and we explain that the adaptive training in our paper does not consume significant training resources.
> - W3: We explain that our work is the first systematic investigation into the impact of adaptive training on deploying detection models to assist MLLMs, which is an innovative contribution to the field.
> - W4.2:  We clarify the reviewer's misunderstanding and explain that all training strategies use the same LLM backbone, specifically Vicuna-1.5.
>
> ---
>
> ## W1: "The key insights may seem to reiterate the existing observations of the related papers."
>
> The focus of these works is different from ours. MiniGPT4-v2, VisionLLM, and Shikra (mentioned in our paper) all enhance their datasets with a large amount of object detection data, as well as **use special tokens such as "<det>"** to perform object detection downstream tasks. **They don't deploy independent detection models during training or inference.** In contrast, our work focus employing off-the-shelf vision detection models for real-time detection information generation, similar to the multi-agent approach.
>
> Another key distinction is that the detection information introduced by MiniGPT4-v2 and VisionLLM is  **completely accurate** since it is derived from datasets. However, in our work, the detection models may occasionally produce errors, **introducing noise that affects the training-free model**.
>
> In short, our research investigates **whether adaptive training can help MLLMs better identify noise in real-time detection information and more effectively leverage the outputs of additional detection models to enhance VQA performance.** This is fundamentally different from the objectives of related works like MiniGPT4-v2 and VisionLLM.
>
> (To be continued.)

---

> ### Author Response · Authors · 2024-11-23
>
> # Responses to Reviewer JV9J [Part 2/3]
> ---
> (Continued from the previous page. W1)
>
> ## W2: "... overlooks the effectiveness of training-free methods."
>
> **We don't overlook the efficiency of training-free methods**. In lines 51-53 and 182-187 of our paper, we clearly state that training-free approaches are simple, resource-efficient, and cost-effective. In our paper, we **objectively conduct a comparative analysis** of the performance of three different training strategies. The conclusion that the performance of training-free models is inferior to that of the trained models is **based on objective experimental results**.
>
> Regarding resource consumption, our fine-tuning strategy uses the visual instruction tuning dataset of MLLMs, which is relatively small in size and does not require excessive training time. Given the significant performance improvement, this approach is highly cost-effective. Additionally, in Section D.1 of the appendix, we demonstrate that fine-tuning Qwen-VL with LLaVA-1.5's small-scale fine-tuning dataset (665K) achieves excellent results. This suggests that fine-tuning does not necessarily require the model's original dataset, and high-quality VQA datasets with small scales are also effective. Furthermore, we try removing the detection data of the LLaVA-1.5 fine-tuning dataset (from 665K to 450K), and the resulting model still performs excellently. The results are presented in the table below, and the corresponding model name is "LLaVA-1.5-7B-FTBI-FNDI". The data size is already very small, and we can further explore reducing the data size to improve efficiency even more.
>
> | Model                      | VQA$^{v2}$ | GQA*     | VQA$^{T}$ | MME$^{P}$  | MME$^{C}$ | POPE     | MMB      | MMB$^{CN}$ | MM-Vet   | SEED     |
> | -------------------------- | ---------- | -------- | --------- | ---------- | --------- | -------- | -------- | ---------- | -------- | -------- |
> | LLaVA-1.5-7B               | 78.5       | 79.6     | 58.2      | 1510.7     | 355.7     | 85.9     | 64.3     | 58.3       | 30.5     | 58.6     |
> | **LLaVA-1.5-7B-TFI**       | 78.5       | 79.2     | 59.2      | 1497.0     | 401.0     | **89.9** | 65.0     | 57.2       | 33.7     | 60.6     |
> | **LLaVA-1.5-7B-FTBI-FNDI** | **79.1**   | 79.8     | 59.5      | **1518.0** | **410.4** | 88.8     | **68.4** | **60.3**   | 33.9     | **61.1** |
> | **LLaVA-1.5-7B-FTBI**      | 79.0       | **80.1** | **60.1**  | 1482.7     | 397.9     | 88.9     | 67.3     | 60.2       | **35.2** | 60.8     |
>
>
>
>
> ---
>
>
> ## W3: "The novelty of the proposed methods is limited..."
>
> Our work focuses on **comparing the performance gains brought by the training-free strategy and the adaptive training strategies for employing detection models to assist MLLMs**, rather than proposing a new method. The training approach we select aligns with the training paradigms of advanced MLLMs. Specifically, the introduction of LoRA modules achieves good results in both the pre-training and fine-tuning stages of MLLM training. This training method is sufficient to enable an objective comparison of different training strategies.
>
> Most research on deploying detection models to assist MLLMs is based on a training-free approach. To our knowledge, there are no exisiting studies that systematically investigate the comparison between the training-free and adaptive training methods. With the aim of inspiring further research in this area, we first explore this aspect. Our work is innovative.
>
> (To be continued.)

---

> ### Author Response · Authors · 2024-11-23
>
> # Responses to Reviewer JV9J [Part 3/3]
> ---
> (Continued from the previous page. W2&W3)
>
> ## W4.1:"Why is the retraining strategy not training visual encoders?"
>
> The retraining strategy refers to training from scratch as advanced MLLMs do, rather than directly retraining off-the-shelf models. All MLLMs must undergo training from scratch, and our retraining strategy simply refer to this process.
>
> In our work, we do not train the visual encoder because the baseline we use, LLaVA-1.5-7B, also keeps the visual encoder frozen during training. To address your question more thoroughly, we conduct additional experiments as requested, where we unfreeze the visual encoder. We repeat both the retraining and fine-tuning processes. The new results are presented as follows, where "TVE" denotes training with the visual encoder unfrozen.
>
> | Model                     | VQA$^{v2}$ | GQA*     | VQA$^{T}$ | MME$^P$    | MME$^C$   | POPE     | MMB      | MMB$^{CN}$ | MM-Vet   | SEED     |
> | ------------------------- | ---------- | -------- | --------- | ---------- | --------- | -------- | -------- | ---------- | -------- | -------- |
> | LLaVA-1.5-7B              | 78.5       | 79.6     | 58.2      | 1510.7     | 355.7     | 85.9     | 64.3     | 58.3       | 30.5     | 58.6     |
> | LLaVA-1.5-7B-TFI          | 78.5       | 79.2     | 59.2      | 1497.0     | 401.0     | **89.9** | 65.0     | 57.2       | 33.7     | 60.6     |
> | **LLaVA-1.5-7B-RBI-TVE**  | 78.2       | 76.1     | 59.3      | 1466.5     | 396.4     | 89.1     | 67.2     | 60.4       | 34.0     | 60.5     |
> | **LLaVA-1.5-7B-FTBI-TVE** | **79**     | **79.7** | **60.4**  | **1556.9** | **412.1** | 89.3     | **68.9** | **61.2**   | **34.6** | **60.8** |
>
>
>
> | Model                            | VQA$^{v2}$ | GQA* | VQA$^{T}$ | MME$^{P}$ | MME$^C$ | POPE | MMB  | MMB$^{CN}$ | MM-Vet | SEED |
> | -------------------------------- | ---------- | ---- | --------- | --------- | ------- | ---- | ---- | ---------- | ------ | ---- |
> | LLaVA-1.5-7B                     | 78.5       | 79.6 | 58.2      | 1510.7    | 355.7   | 85.9 | 64.3 | 58.3       | 30.5   | 58.6 |
> | **LLaVA-1.5-7B-RBI-TVE w/o DI**  | 76.4       | 75.4 | 56.1      | 1480.7    | 289.3   | 83.1 | 66.3 | 59.5       | 30.1   | 59.6 |
> | **LLaVA-1.5-7B-FTBI-TVE w/o DI** | 78.1       | 78.9 | 57.7      | 1499.6    | 318.6   | 85.5 | 66.8 | 60.1       | 30.8   | 60.5 |
>
> As shown in Table 1, even with the visual encoder being trained, the performance of the training-free, retraining, and fine-tuning strategies **aligns with the patterns summarized in our paper**. Specifically, the RBI model outperforms the training-free model, while the FTBI model further surpasses the RBI model. Moreover, the fine-tuned model achieves the best performance in 9 out of 10 benchmarks while training with the visual encoder unfrozen.
>
> Furthermore, Table 2 presents the performance of RBI and FTBI models when the detection information is not dynamically incorporated during inference. It demonstrates that, under the condition where the visual encoder is unfrozen, the fine-tuned model still maintains comparable performance to the original LLaVA-1.5-7B, while the RBI model performs worse than the original model. This indicates that the fine-tuning strategy better balances the contributions of the image encoder's outputs and the detection information, thereby facilitating a more effective understanding of detection cues. These findings are consistent with the conclusions presented in our paper.
>
> Thank you for your valuable comment. We have included this discussion in Appendix D.5 of our revised paper.
>
> ---
>
> ## W4.2: "... the different backbone is used for retraining"
>
> We note a misunderstanding by reviewer regarding this point. **Both the retraining strategy and the fine-tuning strategy utilize the Vicuna-1.5 backbone.** The term "LLaVA-1.5" in the figure refers to the Vicuna-1.5 architecture as well, as the LLM backbone of LLaVA-1.5 is also based on Vicuna-1.5.
>
> We distinguish between "Vicuna-1.5" and "LLaVA-1.5" in the figure for clarity:
>
> - For the retraining strategy, the LLM backbone is initialized with the original weights of Vicuna-1.5. Because we need to replicate the pretraining and fine-tuning stages of LLaVA-1.5 from scratch.
> - For the training-free and fine-tuning strategies, "LLaVA-1.5" represents that we use the Vicuna-1.5 weights derived from LLaVA-1.5 to initialize the LLM backbone, which have already been fine-tuned on the LLaVA-1.5 dataset.
>
> We have detailed this distinction in our paper, specifically in lines 258–279.
>
>
>
> ---
>
> ## Closing Remarks
>
> We would like to express our sincere gratitude for your suggestions and queries. We hope that our responses have addressed your concern and that you will consider re-evaluating our paper. Please do not hesitate to contact us if you have any further inquiries or recommendations. We appreciate your time and feedback immensely.

---

> > ### Comment · Reviewer_JV9J · 2024-11-26
> >
> > Thanks for the detailed responses. Almost all of my concerns were resolved, so I raised my score to 5.
> > One reason for why I could not give a higher score is the novelty from the previous detection-infused models such as MiniGPT-v2, VisionLLM which generate the same detection information. The motivation behind why we should focus on using off-the-shelf detection models rather than choosing these widely adopted baselines might need further elaboration and analysis.

---

> ### Author Response · Authors · 2024-11-27
>
> Dear Reviewer JV9J,
>
> Thank you very much for taking the time to read our responses and re-evaluate our paper! We are sincerely grateful for your decision to increase your rating!
>
> We notice that the reviewer still have concerns regarding *“the motivation behind why we should focus on using off-the-shelf detection models rather than choosing the widely adopted baselines.”* Below, we provide a detailed explanation to address this point, and we hope it will help clarify the reviewer's concerns.
>
> ---
>
> `
> 1.No Absolute Superiority Between the Two Paradigms:
> `
>
> - First and foremost, we want to clarify that **we do not claim that** deploying detection models to assist MLLMs is inherently better than choosing those widely adopted baselines. Instead, our paper aims to conduct systematic comparative experiments based on the former paradigm, fostering the understanding of the impact of adaptive training on its performance.
> - Whether by deploying off-the-shelf detection models or by introducing special tokens for object detection downstream tasks, **both approaches enhance the object detection capabilities of MLLMs** and enable them to better focus on detailed information within images. Specifically, if we integrate special tokens into object detection data during MLLM training, as done in MiniGPT-v2 and VisionLLM, the MLLM itself could learn to understand these special tokens and follow their guidance to perform end-to-end object detection reasoning. Alternatively, deploying independent detection models to generate real-time detection information as input could enable MLLMs to directly access image details through context-enhanced text. **Each approach has its unique strengths, and neither is inherently superior to the other.**
>
> ---
>
> `
> 2.The Role of Detection Information Differs Between the Two Paradigms:
> `
>
> While both paradigms involve detection information, the role of this information differs significantly:
>
> - The method of deploying detection models allows MLLMs to receive real-time detection information during both training and inference. This type of detection information encompasses the locations of all detectable objects in the image, containing rich details about the image.
> - The special token method, which does not deploy detection models, requires manual input at the input stage. Such information is typically limited to a single object or a small number of objects, serving primarily as task guidance. Next, this approach requires the MLLM to output detection information in a formatted manner, **guiding the MLLM to complete the detection task**.
>
> Thus, in the paradigm of deploying detection models, detection information assists MLLMs in downstream tasks by providing useful detection details. In contrast, in the special token paradigm, detection information **usually acts as a signal to indicate that the task involves detecting specific targets and guides the MLLM to finish the detetion task**. While both paradigms infuse detection information, their functions differ: the former serves as an auxiliary function, and the latter acts as an instructive function.
>
> ---
>
> `
> 3.Our Study is a Pioneering Work, Offering Inspiration for Further Research:
> `
>
> - Deploying independent detection models (or models for other downstream tasks) to generate context-enhanced text for assisting MLLMs is straightforward and effective. Furthermore, the deployed models are interchangeable, which provides excellent scalability. Considering the strong points, **an increasing number of researchers are investigating this paradigm and working based on it.**
> - Nevertheless, many researchers tend to adopt training-free strategies. The impact of adaptive training, however, remains an important area of investigation. Therefore, we conduct systematic experiments based on **the training-free and adaptive training strategies in this paradigm, as there has not been a comprehensive comparison between them.**
> - Our findings demonstrate that the adaptive training strategy indeed outperforms the training-free strategy. Additionally, we confirm that fine-tuning with only a small amount of high-quality VQA data can also lead to improved performance, and the performance gain is still preserved even after replacing the detection models. As a pioneering study in this area, **we have uncovered many valuable insights, and we hope our findings will inspire researchers in related fields.**
>
> ---
>
> We newly submit a revision of our paper. We have included these discussions into the Related Works Section and Appendix D.7.
>
> We hope that our responses adequately address your comments and help strengthen your confidence in the acceptance of our paper. If you have any additional concerns or queries, we warmly encourage you to share them with us. Thank you once again for your valuable time and feedback!
>
> Best, authors

---

> ### Author Response · Authors · 2024-12-01
> **Gentle reminder of the author-reviewer discussion deadline**
>
> Dear Reviewer JV9J,
>
> We would like to know if you have any further questions regarding our latest discussion points. We hope that our responses have clarified the misunderstandings you raised. We greatly appreciate the time and effort you have invested in reviewing our paper. Thank you!
>
> Best, authors

---

> ### Author Response · Authors · 2024-12-03
> **Gentle reminder of the author-reviewer discussion deadline**
>
> # New Responses to Official Comment by Reviewer JV9J [Part 1/2]
>
> Dear Reviewer JV9J,
>
> We notice that you have not replied to our latest response. We sincerely hope that our response has addressed your remaining concerns. In addition to the previous response, below, we now provide more examples and references to support our arguments.
>
> ------
>
> `1. Comparison of Detection Information Examples: MiniGPT-v2, VisionLLM, and Ours.  `
>
>  We further compare the detection information examples of MiniGPT-v2, VisionLLM, and our method, providing a more detailed illustration of the significant functional differences between them.
>
>
>
> - **MiniGPT-v2**
>
> The question and answer format of MiniGPT-v2 is similar to the following example. The question is: *"[grounding] please describe this image as detailed as possible,"* and the answer is: *"<p>A crepe</p> {<38><51><90><78>} sits on <p>a white plate</p> {<29><45><100><83>}...".*
>
> MiniGPT-v2 introduces a special token "[grounding]" to indicate that the current task involves object location marking. Additionally, special symbols such as "<p>," "</p>," and "{}" are introduced to construct detection information.
> As shown in the example, the detection information in MiniGPT-v2 **appears in the answer** and is specifically used to **guide the model in completing the "grounding" task** (next token prediction). The MLLM follows this detection information format to output grounding results.
>
>
>
> - **VisionLLM**
>
> The question and answer format of VisionLLM follows a structure similar to the example below. The question is: *"Identify the objects in <image> that belong to {'What is the child eating?': <c0>, 'red gamepad': <c1>} and draw a bounding box around each one...",* while the corresponding answer is: *"The bounding boxes are [(<c0>, 226.4, 229.8, 363.1, 347.4), (<c1>, 441.1, 183.5, 538.6, 269.9)]."*
>
> As shown in the example, VisionLLM introduces special tokens, such as "{}," "<c0>," and "[]," to structure detection information. These tokens are designed to **specify the task**, **guiding the MLLM to output object location information** in the required format (next token prediction).
>
>
>
> - **Ours**
>
> The question and answer format of our method follows a structure similar to the example below. The question is, *"Here are the central coordinates of certain objects in this image: 2 people: {[0.25, 0.12], [0.11, 0.43]}, 1 cake: {[0.42, 0.32]}... What number is on the cake?"* The corresponding answer is, *"The number on the cake is '9'."*
>
> Our detection information consists of **the number and locations of all detectable objects within the image** and is incorporated **into the MLLM's input**. Unlike the detection information in MiniGPT-v2 and VisionLLM, our detection information includes the fine-grained location details of all detectable objects, which serve as auxiliary information to support the reasoning process of MLLMs.
>
>
>
> - **Comparison**
>
>  MiniGPT-v2 and VisionLLM primarily use special tokens to instruct MLLMs on how to **format their outputs as detection information**. In this case, the special tokens and the detection information mainly serve as task guidance, guiding the MLLM to finish the grounding tasks.
>
> In contrast, our approach utilizes detection information to **supplement MLLMs' inputs with detailed visual information from images**. By collecting all detectable objects within an image, we create enhanced contextual content to support and enrich the MLLMs' capabilities.
>
> Therefore, the role of detection information in MiniGPT-v2 and VisionLLM differs from ours, as theirs serves as guidance, while ours functions as supplementary assistance.
>
> (To be continued.)

---

> ### Author Response · Authors · 2024-12-03
> **Gentle reminder of the author-reviewer discussion deadline**
>
> # New Responses to Official Comment by Reviewer JV9J [Part 2/2]
>
> ---
>
> (Continued from the previous page.)
>
>
> `2.  A Further Explanation of the Research Motivation.`
>
> Here, we further explain the motivation behind our current research. First, let us review some recent related studies: GLEE [1] enhances MLLMs' performance by feeding textual object queries into the backbone LLM. P2G [2] and IVE [3] employ detection agents to generate textual grounding clues for improved reasoning. Power-LLaVA [4] utilizes an object detector to produce textual class and location information to assist the MLLM in generating high-quality outputs. VLPrompt [5] leverages an object detector to generate target names and infer relationships, thereby aiding MLLMs in reasoning tasks. While these models all adopt the approach of deploying detection models to generate grounding information for MLLMs, **none of them have explored adaptive training**.
>
> Our work systematically investigates the impact of adaptive training on the performance of the approach that deploys detection models to assist MLLMs. Our findings demonstrate that the adaptive training strategy indeed outperforms the training-free strategy. Moreover, we confirm that fine-tuning with only a small amount of high-quality VQA data can also lead to improved performance, and this performance gain is preserved even after replacing the detection models. We hope that our experimental results will **inspire researchers in related fields to consider introducing appropriate adaptive training to enhance model performance.**
>
>
>
> [1] General object foundation model for images and videos at scale.
>
> [2] Plug-and-play grounding of reasoning in multimodal large language models.
>
> [3] Incorporating Visual Experts to Resolve the Information Loss in Multimodal Large Language Models.
>
> [4] Power-LLaVA: Large Language and Vision Assistant for Power Transmission Line Inspection.
>
> [5] Vlprompt: Vision-language prompting for panoptic scene graph generation.
>
> ------
>
> May we kindly ask if you have any further questions regarding your remaining concern? We would greatly appreciate it if you could provide us with your feedback at your convenience. Thank you very much for taking the time to review and consider our comments.
>
> Best, authors

---

### Official Review · Reviewer_j1zf · 2024-11-04

**Soundness:** 2
**Presentation:** 2
**Contribution:** 2
**Rating:** 6
**Confidence:** 3

**Summary:**

The paper investigates the impact of training strategies on Multimodal Large Language Models (MLLMs) when integrating textual detection information from vision models. While current methods often utilize a training-free approach, the researchers systematically explore the effects of adaptive training, retraining, and fine-tuning strategies. Their findings indicate that fine-tuning significantly enhances the MLLMs' performance—improving results compared to the training-free method across various benchmarks. Additionally, fine-tuning enables MLLMs to retain performance benefits even after replacing detection models, suggesting better comprehension of the specialized textual information.

**Strengths:**

- The paper is well-written and easy to follow.
- Through extensive empirical validation, the study rigorously evaluates the performance of various training strategies across different experimental settings.

**Weaknesses:**

- Limited Contribution
    - The primary findings of this study demonstrate limited novelty in their conclusions. The superiority of fine-tuning over training-free methods has been well-established in the literature, making this result somewhat predictable. Furthermore, the inclusion of comparison with retraining from scratch adds limited value, as it is rarely considered a preferable option in practice.
- Ambiguities in Dataset Construction
    - The proportional distribution of various data types, including textual detection information needs to be more adequately specified. Moreover, the paper's use of the term "infusion" lacks a precise definition, leaving uncertainty about whether it refers to data addition or conversion processes. The paper's ambiguous description of data processing methods is problematic, especially since data conversion, if implemented, would reduce conventional question-answer pairs and potentially affect benchmark performance, particularly in the retraining strategy.

**Questions:**

- How is the textual detection instruction data infused during training? (See weakness)

**Details Of Ethics Concerns:**

The work does not have ethical concerns. Since the framework inherits the limitations of LLMs and MLLMs, the framework may share the concerns of those large foundation models. However, such concerns are not specific to this work.

---

> ### Author Response · Authors · 2024-11-23
>
> # Responses to Reviewer j1zf
> ---
>
> Dear Reviewer j1zf, we sincerely thank you for your time and efforts in reviewing our paper, as well as your feedback for this work! However,  we have carefully read all your comments and believe there may be some misunderstandings regarding our work. We address each of your concerns, hoping to clarify our paper and encourage you to lean toward its acceptance.
>
>
> ---
> ## Foreword
>
> We would like to point out the key misunderstanding here. Our paper is **not** focused on dataset construction, but rather on exploring the integration of detection models—similar to Multi-Agent—into the MLLMs for real-time assistance. We focus on deploying vision detection models, rather than generating detection data. Additionally, we investigate the impact of applying training strategies when the detection models are introduced, as well as demonstrate how the synergy between MLLMs and vision detection models, combined with tailored training techniques, can achieve performance improvements.
>
> ---
>
> ## W1.1: "The superiority of fine-tuning over training-free methods has been well-established in the literature."
>
> Related works on fine-tuning are primarily based on introducing new datasets or modifying the architecture of models. To our best knowledge, there has been **no previous work** systemcally and specifically studying the effect of fine-tuning with incorporated additional detection models.
>
> Existing research on introducing detection models typically considers training-free methods. With that in mind, it's meaningful to propose a work that demonstrates how adaptive training can lead to performance improvements with additional detection models incorporated. Our work can inspire researchers to explore better model performance while they employ vision detection models to assist MLLMs.
>
>
> ---
>
>
> ## W1.2: "The comparison with retraining from scratch adds limited value."
>
> We respectfully disagree with the reviewer's opinion. First, it is important to clarify that the retraining strategy does not imply retraining a pre-trained model; rather, they represent training from scratch as the advanced MLLMs do. **In practice, all MLLMs must undergo training from scratch, and our retraining strategy simply refer to this process.** Therefore, such a comparison is highly relevant to practice.
>
> Furthermore, without incorporating a comparison with the retraining strategy, we would miss several meaningful insights presented in our paper. For instance, we demonstrate that first training an MLLM without the detection models infused, followed by fine-tuning it with the detection models infused, enables a better balance between the image encoder's output and the detection information. This valuable finding is derived from the comparison with the retraining strategy.
>
> Thus, the comparison of the retraining strategy is valuable as it can help us understand the practical implications of adaptive training.
>
>
> ---
>
>
> ## W2: Ambiguities in Dataset Construction
>
> We would like to reiterate that **our work does not focus on dataset contributions**. The paper does not introduce any new datasets, and therefore, there is no discussion of data processing methods or dataset construction. Instead, our work introduces additional detection models (similar to Multi-Agent), which generates real-time detection information to assist MLLMs in reasoning.
>
> Regarding the format of the detection information derived from the detection models, it follows related Multi-Agent or MLLM works, such as P2G [1], Power-LLaVA [2], and CogVLM [3]. The experimental results in these papers have already validated the effectiveness of the format as "object + coordinates". Moreover, as demonstrated by the experiments in our paper, this approach is indeed effective.
>
> Furthermore, we have provided a clear explanation of the "infusion" strategy within our paper. For instance, Figure 2 (lines 216-227) clearly illustrates how the textual detection information passes through the LLM's embedding layer before being concatenated with the image features at the embedding level. Additionally, Section 3.2 (lines 245-254) provides a detailed and specific description of the infusion process.
>
>
>
> ---
>
> [1] Plug-and-play grounding of reasoning in multimodal large language models.
>
> [2] Power-llava: Large language and vision assistant for power transmission line inspection.
>
> [3] CogVLM: Visual Expert for Pretrained Language Models
>
> ---
>
> ## Closing Remarks
>
> We sincerely appreciate the valuable time you have spent in reviewing our paper.  We respectfully ask you to take another look at the merits of our work. We hope our responses have addressed your comments and can enhance your confidence in the acceptance of this paper. If you have any additional concerns or queries, we warmly invite you to share them with us. Thanks again!

---

> > ### Comment · Reviewer_j1zf · 2024-11-30
> >
> > First of all, sorry for the late response.
> > I have carefully re-read the paper and the reviews from other reviewers. I still have two questions about the dataset part, which I believe could be easily clarified:
> >
> > - **Dataset Construction:**
> >     - I understand that dataset construction is not the main contribution of this work. I may have overlooked that the authors use "the same instruction tuning data as the original MLLMs." To clarify, does the infusion process apply to all the data in the original dataset by extracting detected objects and texts from OCR?
> > - **Distribution of Data:**
> >     - In addition to the analysis of the "length" of detection information in Table 5, what is the ratio of the data that contains text (OCR) data among all of the datasets? I believe it is important to be aware of this ratio to assess the evaluation process and interpret the results in depth.

---

> > > ### Author Response · Authors · 2024-12-01
> > >
> > > Thank you very much for your response to our replies! We notice that you raise two additional questions. We will address them in detail as follows:
> > >
> > > ---
> > >
> > > `
> > > 1."Does the infusion process apply to all the data in the original dataset?"
> > > `
> > >
> > > Yes, we do employ detection models to perform object detection and text recognition **for all data containing images**. This process is seamlessly integrated into the training phase, where, at each training step, **every sample undergoes detection through the deployed detection models**.
> > >
> > > During the training phase, the processed detection information is passed through the Text Embedding Layers, where the corresponding textual features are extracted. These features are then concatenated with the image features and fed into the MLLMs, effectively enhancing their reasoning capabilities.
> > >
> > > Additionally, we ensure that **OCR is performed on all images, regardless of whether they contain text.** If the OCR results indicate that no text is present, we will not generate OCR-related detection information and only provide detection information related to object detection.
> > >
> > > ---
> > >
> > > `
> > > 2."What is the ratio of the data that contains text?"
> > > `
> > >
> > > To address the question you raise, **we conduct an additional experiment and deploy the PaddleOCRv2 to analyze the number of text-containing images** within LLaVA's original dataset. The results show that, among the 624K image samples, **197K images contain text, which accounts for 31.56%**.
> > >
> > > Regarding the analysis of Table 5, we initially stated that "about 0.2% of OCR information surpasses the 512 threshold." Upon review, we realize that this is an oversight, and we sincerely appreciate your attention to this matter. **We will definitely revise this statement in our final version, as "about 0.67% of text-containing images generate OCR information that surpasses the 512 threshold."** We are grateful for your valuable feedback.
> > >
> > > Furthermore, we conduct a deeper analysis of the length of OCR information. The data shows that 0.67% of text-containing images generate OCR information exceeding the 512 threshold, which remains a relatively small proportion. Besides, we observe that when excluding images without text, the average length of OCR information is 97.5, which is also quite small. These results further **corroborate the effectiveness of the studied compression strategy**, ensuring that the vast majority of the detection information remains within reasonable limits.
> > >
> > > ---
> > >
> > > We hope that our explanations adequately address your questions and may lead to a higher evaluation of our paper. We greatly appreciate the time and effort you have invested in reviewing our paper. If you have any further questions or require additional clarifications, we warmly welcome you to share them with us. Thank you!
> > >
> > > Best, authors

---

> > > > ### Comment · Reviewer_j1zf · 2024-12-02
> > > >
> > > > Thank you for your response. As all my concerns are addressed, I raise my rating to 6.

---

### Official Review · Reviewer_Excq · 2024-11-04

**Soundness:** 3
**Presentation:** 3
**Contribution:** 3
**Rating:** 6
**Confidence:** 4

**Summary:**

The paper  investigates the impact of various training strategies on the multimodal large language models' (MLLMs) ability to utilize infused detection information. The authors propose three training strategies—training-free infusion, retraining, and fine-tuning—for incorporating detection data in textual format. Through extensive experimentation across benchmarks, they conclude that fine-tuning yields the best results, enhancing MLLM performance in tasks requiring fine-grained image recognition by up to 6.71% over training-free methods. The study also explores model adaptability to different detection models, suggesting fine-tuning as a robust approach for integrating detection information.

**Strengths:**

1. The paper addresses a crucial aspect of MLLMs' limitations—difficulty in interpreting detailed visual elements. By exploring methods to effectively integrate detection information, it has significant implications for real-world applications where precision in visual recognition is essential, such as autonomous driving, medical imaging, and other fields that rely on high-detail visual data.
2. The authors conduct a wide-ranging analysis across ten well-regarded benchmarks, providing robust evidence for the effectiveness of each training strategy.
3. A key strength is the demonstration of fine-tuning’s adaptability when incorporating different detection models. The authors showcase that fine-tuned models retain performance gains even when switching from closed-set to open-set detectors, underscoring fine-tuning as a resilient strategy for enhancing MLLMs.
4. The findings from comparing training-free, retraining, and fine-tuning strategies offer valuable empirical insights. By quantitatively showing the superiority of fine-tuning, the paper guides future work on the practical application of training strategies for MLLMs that require fine-grained detail recognition.

**Weaknesses:**

1. Fine-tuning MLLMs with detection information likely introduces computational overhead, which is not sufficiently addressed. An analysis of training costs and memory requirements across the three strategies would provide valuable insights into the feasibility of each approach for large-scale applications.
2. While the paper includes multiple benchmarks focused on fine-grained visual tasks, the evaluation could benefit from additional benchmarks that test broader language-vision capabilities. Tasks like DocumentVQA.
3. The paper does not examine how variations in detection model accuracy (e.g., OCR quality) impact the MLLM’s performance. Given that the approach depends on external detection outputs, this vulnerability could lead to inconsistent performance if detection quality fluctuates across different scenarios or datasets.

**Questions:**

1. Could you elaborate on the computational requirements for each training strategy, particularly the memory and time costs associated with fine-tuning compared to the training-free approach?
2. Is there a risk of the model overfitting to the textual detection information during fine-tuning? Has the paper examined the impact of fine-tuning on tasks unrelated to detection, to confirm that broader language comprehension capabilities are maintained?

---

> ### Author Response · Authors · 2024-11-23
>
> # Responses to Reviewer Excq [Part 1/4]
> ---
>
> Dear Reviewer Excq, we greatly appreciate your recognition of the practical relevance and the robust evaluation of our work! Thank you for your constructive and detailed review. We make the following detailed responses point by point to address your comments with substantial experimental support. We hope the responses will convince you to lean more toward the acceptance of our paper.
>
>
> ---
> ## Response Overview
>
> We first summarize our response and show more details below.
>
> 1. **More Experiments**
>
> - W2: We evaluate our models on two DocumentVQA benchmarks, DocVQA and InfographicVQA.
> - W3: We conduct experiments based on YOLOv5N and YOLOv11L to investigate the impact of detection model accuracy on MLLM performance.
> - Q2.2:  We remove the detection data from the instruction tuning dataset and repeat the FTBI experiment, aiming to investigate whether the model can still maintain a broad language comprehension capability.
>
> 2.  **Clarification of Certain Aspects of Our Paper**
>
> - W1 & Q1: We have already discussed the training cost in our paper. And we add a statement of the memory usage.
> - Q2.1: We explain that the discussion of model overfitting is already in our paper.
>
> ---
>
> ## W1 & Q1: Time Consumption and Memory Requirements
>
> We have already detailed the training costs in Appendix B.6. The reviewer may have overlooked it. Specifically, on four A100 GPUs (80GB), the time consumption is as follow:
>
> - For pretraining LLaVA-1.5-7B, the time increases from 6 hours to 11 hours.
> - For pretraining LLaVA-1.5-13B, the time increases from 11 hours to 17 hours.
> - For fine-tuning LLaVA-1.5-7B, the time increases from 16 hours to 22 hours.
> - For fine-tuning LLaVA-1.5-13B, the time increases from 26 hours to 33 hours.
>
> Regarding the memory requirements, deploying detection models results in an additional GPU memory usage of up to 4GB in each GPU compared to not deploying detection models.
>
> Considering the performance improvements achieved, these time costs and memory costs are relatively minor and cost-effective. Moreover, researchers can opt for detection models with faster inference speeds, lower memory usage, and higher efficiency, further reducing the resource consumption and improving model performance.
>
> We have made these details clearer in the revised version of our paper and marked them in red.
>
> (To be continued.)

---

> ### Author Response · Authors · 2024-11-23
>
> # Responses to Reviewer Excq [Part 2/4]
> ---
> (Continued from the previous page. W1&Q1)
>
> ## W2: Evaluations on DocumentVQA
>
> Following your suggestion, we evaluate our models on two well-known DocumentVQA benchmarks, DocVQA and InfographicVQA. The new results are presented in the two tables below. The first table compares the performance of the TFI, RBI, and FTBI models on the DocVQA and InfographicVQA benchmarks. The second table compares the performance of the RBI and FTBI models on the same benchmarks without inputting detection information during inference.
>
> | Model                  | DocVQA | InfographicVQA |
> | ---------------------- | ------ | -------------- |
> | LLaVA-1.5-7B           | 19.4   | 18.8           |
> | **LLaVA-1.5-7B-TFI**   | 35.3   | 21.0           |
> | **LLaVA-1.5-7B-RBI**   | 35.7   | 20.9           |
> | **LLaVA-1.5-7B-FTBI**  | 35.9   | 21.3           |
> |                        |        |                |
> | LLaVA-1.5-13B          | 20.6   | 20.7           |
> | **LLaVA-1.5-13B-TFI**  | 35.5   | 22.1           |
> | **LLaVA-1.5-13B-RBI**  | 37.9   | 23.3           |
> | **LLaVA-1.5-13B-FTBI** | 38.5   | 24.2           |
>
> As shown in Table 1, the deployment of detection models, particularly the OCR model, leads to a significant score improvement on DocVQA. Furthermore, models with adaptive training noticeably outperform training-free models . Specifically, the FTBI models surpass the RBI models, which in turn outperforms the TFI models. This suggests that the adaptive training enables MLLMs to better leverage the input detection information, resulting in improved performance.
>
> | Model                         | DocVQA | InfographicVQA |
> | ----------------------------- | ------ | -------------- |
> | LLaVA-1.5-7B                  | 19.4   | 18.8           |
> | **LLaVA-1.5-7B-RBI w/o DI**   | 17.3   | 17.8           |
> | **LLaVA-1.5-7B-FTBI w/o DI**  | 19.4   | 18.7           |
> |                               |        |                |
> | LLaVA-1.5-13B                 | 20.6   | 20.7           |
> | **LLaVA-1.5-13B-RBI w/o DI**  | 18.6   | 20.1           |
> | **LLaVA-1.5-13B-FTBI w/o DI** | 20.6   | 20.9           |
>
> Table 2 presents a comparison between the RBI models and the FTBI models in the absence of textual detection information. As shown, the performance of the RBI models is significantly inferior to that of the FTBI models. While the FTBI models, without detection information, perform similarly to the original LLaVA-1.5. This demonstrates that the fine-tuning strategy allows MLLMs to effectively balance the weights between the image encoder output and textual detection information, thereby preserving the comprehensive VQA capabilities. These results are consistent with the findings in our paper.
>
> Thank you for your valuable suggestion! Incorporating the DocumentVQA benchmarks enhances the validation of our experimental conclusions. We have included this point in Appendix E.3 of the revised paper.
>
> (To be continued.)

---

> ### Author Response · Authors · 2024-11-23
>
> # Responses to Reviewer Excq [Part 3/4]
> ---
> (Continued from the previous page. W2)
>
> ## W3: Impact of Detection Models with Varying Accuracy
>
> The outputs of low-performance detection models often include noise, which can adversely affect the following MLLM. Following your suggestion, we respectively employ a low-performance detection model YOLOv5N and a high-performance detection model YOLOv11L (replacing only the object detection model DINO while keeping the PaddleOCR unchanged) and conduct both the training-free and fine-tuning experiments again. Next, we compare the adaptability of the training-free and fine-tuning strategies.
>
>
>
> | Model                          | VQA$^{V2}$ | GQA*     | VQA$^{T}$ | MME$^{P}$  | MME$^{C}$ | POPE     | MMB      | MMB$^{CN}$ | MM-Vet   | SEED     |
> | ------------------------------ | ---------- | -------- | --------- | ---------- | --------- | -------- | -------- | ---------- | -------- | -------- |
> | **LLaVA-1.5-7B**               | 78.5       | 79.6     | 58.2      | **1510.7** | 355.7     | 85.9     | 64.3     | 58.3       | 30.5     | 58.6     |
> | **LLaVA-1.5-7B-YOLOv5N-TFI**   | 78.3       | 79.3     | 59.0      | 1459.9     | 382.9     | 86.3     | 64.2     | 56.3       | 32.2     | 59.9     |
> | **LLaVA-1.5-7B-YOLOv5N-FTBI**  | 78.6       | 79.9     | 60.0      | 1492.7     | 402.1     | 87.1     | 68.9     | 62.5       | 33.5     | 60.4     |
> | **LLaVA-1.5-7B-YOLOv11L-TFI**  | 78.5       | 79.5     | 59.0      | 1490.6     | 364.6     | 87.9     | 64.7     | 56.5       | 33.8     | 60.3     |
> | **LLaVA-1.5-7B-YOLOv11L-FTBI** | **79.0**   | **80.0** | **60.2**  | 1497.5     | **405.4** | **88.9** | **70.3** | **62.9**   | **34.6** | **60.6** |
>
> The new results are presented in the table, from which the following conclusions can be drawn:
>
> - Under the training-free strategy, for general VQA capabilities, YOLOv5N introduces noise to LLaVA-1.5-7B, resulting in performance degradations. In contrast, YOLOv11L, due to its superior performance, introduces minimal noise and thus has little negative impact.
> - Regarding object detection-related capabilities (POPE & MM-Vet), both YOLOv5N and YOLOv11L bring performance improvements under the training-free strategy. However, the improvement from YOLOv5N is noticeably smaller than that from YOLOv11L, which can be attributed to the difference in model performance. This suggests that the training-free strategy exhibits poor adaptability to low-performance detection models.
> - Furthermore, after fine-tuning, both two versions of the MLLM achieve comprehensive performance improvements, surpassing the original LLaVA-1.5-7B. The results align with the conclusions in our paper, demonstrating that the fine-tuning strategy enables the MLLM to better differentiate between noise and useful information and more effectively interpret specially designed detection information, leading to performance enhancement.
>
> These results indicate that **the fine-tuning strategy is more robust and better able to handle the erroneous information** introduced by low-performance detection models compared to the training-free strategy. This further supports our experimental conclusions. We have included this additional experiment in Appendix D.3 of our revised paper. Thank you for your suggestion.
>
>
>
> ---
>
> ## Q2.1: "Is there a risk of overfitting?"
>
> In our paper, we have already included a discussion on model overfitting, which is in Section 4.3 & 4.4. As shown in Table 2 in our paper, models under retraining strategy tend to overfit. These MLLMs overly focus on textual detection information, neglecting the image encoder's output, which leads to a decline in performance on comprehensive VQA benchmarks. Furthermore, as demonstrated in Table 3, the models under fine-tuning strategy do not exhibit overfitting. They strike a good balance between textual detection information and the image encoder outputs, ensuring high performance on both general VQA and object detection benchmarks.
>
> (To be continued.)

---

> ### Author Response · Authors · 2024-11-23
>
> # Responses to Reviewer Excq [Part 4/4]
> ---
> (Continued from the previous page. W3&Q2.1)
>
> ## Q2.2: Fine-tuning on Tasks Unrelated to Detection
>
> Following your suggestion, we conduct fine-tuning experiments using data unrelated to detection tasks, and investigate whether the FTBI model can still maintain broader language understanding capabilities under this training configuration. Regarding the new fine-tuning dataset, we remove samples related to "coordinate" questions (object detection samples) and eliminate all text recognition samples from the original LLaVA fine-tuning dataset. Consequently, the number of samples decreases from 665K to 450K. The new experimental results are presented in the table below, and the corresponding model name is "LLaVA-1.5-7B-FTBI-FNDI".
>
> | Model                      | VQA$^{V}$ | GQA*     | VQA$^{T}$ | MME$^P$    | MME$^C$   | POPE     | MMB      | MMB$^{CN}$ | MM-Vet   | SEED     |
> | -------------------------- | --------- | -------- | --------- | ---------- | --------- | -------- | -------- | ---------- | -------- | -------- |
> | LLaVA-1.5-7B               | 78.5      | 79.6     | 58.2      | 1510.7     | 355.7     | 85.9     | 64.3     | 58.3       | 30.5     | 58.6     |
> | **LLaVA-1.5-7B-TFI**       | 78.5      | 79.2     | 59.2      | 1497.0     | 401.0     | **89.9** | 65.0     | 57.2       | 33.7     | 60.6     |
> | **LLaVA-1.5-7B-FTBI-FNDI** | **79.1**  | 79.8     | 59.5      | **1518.0** | **410.4** | 88.8     | **68.4** | **60.3**   | 33.9     | **61.1** |
> | **LLaVA-1.5-7B-FTBI**      | 79.0      | **80.1** | **60.1**  | 1482.7     | 397.9     | 88.9     | 67.3     | 60.2       | **35.2** | 60.8     |
>
> From the table, it is evident that even without fine-tuning on detection-related data, the FTBI model still demonstrates strong performance, significantly surpassing the original model and the training-free model. Moreover, its results are only slightly below the version fine-tuned with detection data.  These results indicate that, even without fine-tuning on tasks related to detection, the fine-tuned model is still capable of maintaining a broad range of language understanding abilities.
>
> Thank you for your suggestion. We have included this experiment in Appendix D.4 of our revised paper.
>
> ---
>
> ## Closing Remarks
>
> We want to express our deepest gratitude for the constructive suggestions and queries you provide. We dedicate a lot of time and effort to conducting additional experiments based on your recommendations, and we hope that you can consider an increase in the rating if these response with new results effectively address your comments. Thank you!

---

> ### Author Response · Authors · 2024-12-01
> **Gentle reminder of the author-reviewer discussion deadline**
>
> Dear Reviewer Excq,
>
> The discussion period will end in just a few days, and we truly appreciate this valuable opportunity to engage with you.
>
> We notice that you haven't responded to our replies. We believe that your insightful comments, combined with the additional experiments we newly conduct, have helped us further improve our paper. We hope that the new results and responses effectively address your comments, and we kindly ask you to consider a potential increase in your rating. If you have any additional questions or concerns, please feel free to reach out to us. Thank you!
>
> Best, authors

---

### Author Response · Authors · 2024-11-26
**Gentle reminder of the author-reviewer discussion deadline**

Dear Reviewers,

We sincerely thank you for your time and efforts in reviewing our paper, as well as the appreciation and helpful feedback for this work! We carefully check and responded to all your comments, which can be summarized and have been responsed as follows:

`
More experiments with more possible settings:
`

1. We conduct experiments based on YOLOv5N and YOLOv11L to investigate **the impact of detection model accuracy** on MLLM performance. (Reviewer Excq & YDqj) The results show that high-performance detection models can better enhance MLLM performance. Moreover, adaptive training, compared to a training-free approach, enables MLLMs to better adapt to different detection models and effectively mitigate noise.
2. We **unfreeze the visual encoder** and conduct retraining and fine-tuning experiments (Reviewer JV9J), which yields results consistent with the conclusions in our paper.
3. We **remove the detection data from the instruction tuning dataset** and repeat the FTBI experiment (Reviewer Excq & JV9J).  The results show that even with removing detection data during fine-tuning, the model can still maintain broad language comprehension capabilities.
4. We conduct experiments based on LLaVA-NeXT and validate that our conclusions remain effective **on a stronger MLLM** (Reviewer YDqj).
5. We conduct experiments based on Co-DETR-LVIS to explore **the impact of a broader object detection scope** on MLLM performance (Reviewer YDqj).  The results show that expanding the scope introduces redundant information, which does not lead to performance improvements.
6. We evaluate our models **on two DocumentVQA benchmarks**, DocVQA and InfographicVQA  (Reviewer Excq), and achieve results similar to OCR-related benchmarks reported in our paper.

`
Clarification of the Reviewer's Misunderstanding:
`

1. We explain **the core focus of our paper** and compare our work with related works to highlight why **our focus is fundamentally different from theirs** (Reviewer j1zf & JV9J).
2. We explain why our work is **innovative** and how it can provide inspiration to researchers in the relevant field (Reviewer JV9J).
3. We clarify that our focus is to **provide an impartial comparison between the training-free method and adaptive training methods**, without undermining the value of the training-free method (Reviewer JV9J).
4. We clarify that **analyzing and comparing the retraining strategy is valuable**, as it provides insights into the practical implications of adaptive training (Reviewer j1zf).
5. We clarify the reviewer's misunderstanding and explain that **all training strategies use the same LLM backbone**, specifically Vicuna-1.5 (Reviewer JV9J).

The discussion phase will end in a few days and we have received only one response. We believe the paper has been further improved with your helpful comments, and hope that these responses can address your comments.

We respectfully request that you take another look at our responses and the merits of our work. Should you have further questions or require clarification, we warmly welcome your input. Your feedback is highly anticipated and greatly valued. Thanks again!



Best, authors

---

### Meta-Review · Area_Chair_yDUP · 2024-12-20

**Metareview:**

Multimodal Large-Scale Language Models (MLLLMs) are expected to improve accuracy by injecting information from visual recognition models into text format, but the effects of adaptive training have not been fully investigated. This paper examines the impact of injected information on comprehension through comparative experiments with no training, retraining, and fine-tuning strategies. This paper also examines the effects of training on the original capabilities of MLLMs and their compatibility with detection models.

The strengths of this paper are The paper investigates the impact of different training strategies on the performance of a multimodal large-scale language model (MLLM) using injected detection information. It demonstrates the adaptability of fine-tuning when incorporating different detection models. In this paper, a comprehensive set of experiments has been conducted with a thorough analysis of the integration of text detection information into MLLM.

The weakness of this paper is as follows. Although the systematic and concrete investigation of the effects of fine-tuning incorporating additional detection models is novel, the conclusions reached are not particularly innovative.

This paper is rated as borderline. As mentioned above, the academic significance of this paper is recognized for its systematic and specific investigation of the effects of fine-tuning by incorporating additional detection models. On the other hand, the impact of the conclusion that the fine-tuning obtained in the end is superior to the method without training is not significant, and it is difficult to say that it is a strong factor in overcoming the ICLR acceptance bar, so the AC judges it to be rejected at this point. The AC recommends that the paper be submitted to the next conference after considering the reviewer's comments.

**Additional Comments On Reviewer Discussion:**

There were many comments and discussions about the need for experiments in a wider variety of situations, the differences from other research, and the novelty of this research. As a result, three reviewers gave a positive rating and one reviewer gave a negative rating. Therefore, this paper is borderline. As mentioned in the meta-review, the novelty of this paper is that it systematically and concretely investigates the effects of fine-tuning by incorporating additional detection models. On the other hand, the final conclusion is somewhat predictable, and this is the only drawback.

---

### Decision · Program_Chairs · 2025-01-22

Reject